# Lawma: The Power of Specialization for Legal Annotation

**Ricardo Dominguez-Olmedo**[1]**, Vedant Nanda**[2]**, Rediet Abebe**[*,1,3]**,
Stefan Bechtold**[*,4]**, Christoph Engel**[*,5]**, Jens Frankenreiter**[*,6]**, Krishna Gummadi**[*,2]**,
Moritz Hardt**[*,1]**, and Michael Livermore**[*,7]

[1]Max Planck Institute for Intelligent Systems, Tübingen, and Tübingen AI Center  [2]Max Planck Institute for Software Systems, Saarbrücken  [3]ELLIS Institute, Tübingen  [4]ETH Zurich  [5]Max Planck Institute for Research on Collective Goods, Bonn  [6]Washington University in St. Louis - School of Law  [7]University of Virginia School of Law

## Abstract

Annotation and classification of legal text are central components of empirical legal research. Traditionally, these tasks are often delegated to trained research assistants. Motivated by the advances in language modeling, empirical legal scholars are increasingly turning to commercial models, hoping that it will alleviate the significant cost of human annotation. In this work, we present a comprehensive analysis of large language models' current abilities to perform legal annotation tasks. To do so, we construct `CaselawQA`, a benchmark comprising 260 legal text classification tasks, nearly all new to the machine learning community. We demonstrate that commercial models, such as GPT-4.5 and Claude 3.7 Sonnet, achieve non-trivial accuracy but generally fall short of the performance required for legal work. We then demonstrate that small, lightly fine-tuned models vastly outperform commercial models. A few dozen to a few hundred labeled examples are usually enough to achieve higher accuracy. Our work points to a viable alternative to the predominant practice of prompting commercial models. For concrete legal annotation tasks with some available labeled data, researchers are likely better off using a fine-tuned open-source model. Code, datasets, and fine-tuned models are available at https://github.com/socialfoundations/lawma.

## 1 Introduction

The legal system generates a staggering volume of complex documents. United States federal courts alone process hundreds of thousands of cases a year, each having substantial case files. Much empirical legal research involves the systematic collection and analysis of such data in order to understand how laws function in practice and what impact they have on society. What limits researchers across the board is the cost of annotating and classifying legal documents. Legal classification tasks vary in complexity, but often require substantial expertise and effort. Employing trained research assistants stretches to a few thousand documents at a time, but is no match for the sheer scale of legal data.

There has long been an interest by empirical legal scholars in NLP tools for feature extraction (i.e., annotation) in lieu of human annotators (Livermore & Rockmore, 2019). Starting from sentiment analysis and topic models, to now large language models. The costs and error of existing methods is the single most important bottleneck in the empirical legal studies pipeline. Yet, the use of large language models to annotate legal text remains a critically understudied area.

Nonetheless, motivated by the rapid advances in language models, law scholars increasingly try out commercial models, such as GPT-4, on a variety of legal tasks, hoping to boost the efficiency of legal research. The underlying assumption is that large models such as GPT-4 provide the best solution to the problem that is currently available. In this work, we critically examine this assumption.

---

[*]Alphabetical order.

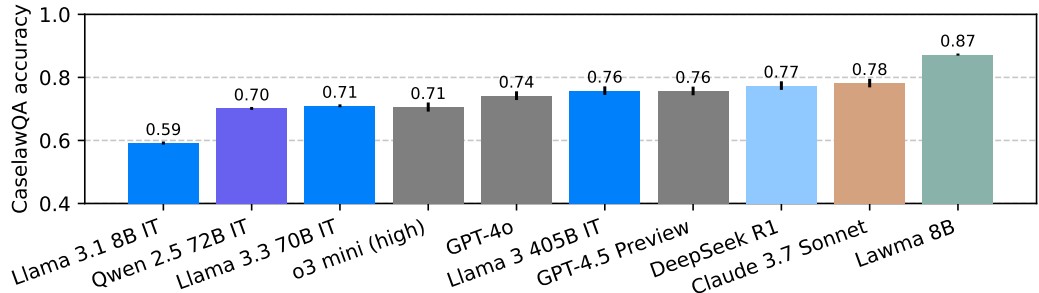

Figure 1: The cost of generality: Performance of various language models on the CaselawQA benchmark for legal annotation. Lawma 8B, specialized for legal annotation, outperforms all other models.

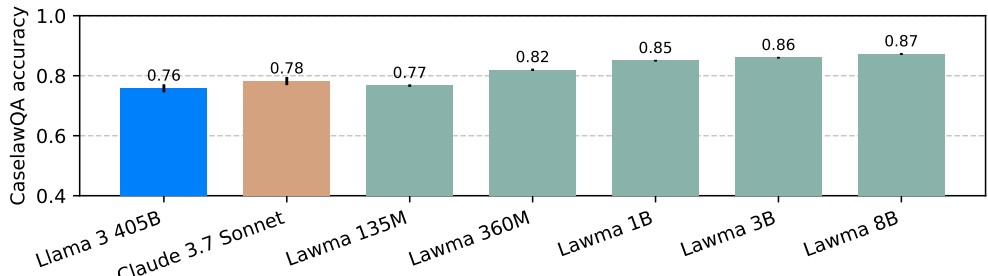

Figure 2: Performance of the Lawma models. The smallest Lawma model, Lawma 135M, is competitive with the best-performing commercial model, Claude 3.7 Sonnet.

## 1.1 OUR CONTRIBUTIONS

We introduce and study a collection of 260 legal classification tasks, nearly all new to the machine learning community. The tasks we introduce are actual legal annotation tasks based on the U.S. Supreme Court (Spaeth et al., 2023) and Court of Appeals (Songer) databases. These databases offer rich annotations for court cases, which we utilize as labels to create challenging multi-class classification tasks. We aggregate these tasks into an easy-to-use benchmark, which we call `CaselawQA`. We detail in Section 2 the process used to construct this benchmark.

Our primary finding is that small, fine-tuned models substantially outperform large commercial models (Figure 1). Specifically, we fine-tune a series of small language models, ranging from 135M to 8B parameters, which we collectively refer to as the `Lawma` models. Our Lawma 8B model achieves **87%** accuracy on CaselawQA, outperforming all commercial models by at least **9** percentage points, with the best-performing commercial model, Claude 3.7 Sonnet, attaining **78%** accuracy. Although it is expected that fine-tuning helps, the superiority of fine-tuning an open-weights model at a much smaller scale is surprising. After all, commercial models are orders of magnitude larger. Our results demonstrate that, for legal annotation, researchers are better off using small specialized models rather than large general-purpose LLMs.

We conduct various large-scale fine-tuning experiments that further demonstrate the benefits and practicality of specializing models for legal annotation:

- Larger models respond better to fine-tuning than smaller models. Accuracy of the Lawma models increases steadily with model size (Figure 2). However, we observe signs of diminishing returns. This suggests that, in the future, major improvements may not come from model scale alone.

- Fine-tuning is data efficient. A few hundred examples typically suffice to achieve higher accuracy than commercial models (Section 4.2, Figure 10). This is crucial, since labeling a few hundred data points is often financially feasible for many legal scholars, whereas labeling many thousands may not.

- Fine-tuning generalizes to unseen tasks. Fine-tuning Llama 3 8B Inst *only* on the Court of Appeals tasks improves its average accuracy on Supreme Court tasks by 18.8 accuracy points (Appendix 4.3, Figure 11).

- We can simultaneously fine-tune on all 260 tasks. There is not a large loss compared with fine-tuning on a specific task (Section D, Figure 13). This is desirable in practice, as it obviates the need to train and maintain a separate model for each task.

- We contextualize our accuracy numbers with intercoder agreement rates. Our analysis reveals task heterogeneity in the relationship between model accuracy and intercoder agreement (Appendix C).

Our results speak to the power of specialization for legal annotation. Our insights suggest that the empirical legal community should invest in an ecosystem of fine-tuned models for relevant annotation tasks. Such an ecosystem could radically expand the capacity of legal scholars to engage in quantitative work.

From a benchmarking perspective, the tasks presented in this work are of independent interest. They are challenging multi-class classification problems that require some amount of legal expertise. The best models achieve non-trivial, but modest performance. And even fine-tuned models don't reach intercoder agreement rates. These legal classification tasks are diverse, non-trivial evaluation tasks for future model advances.

Finally, our work challenges the prevailing narrative about the suitability of "generalist" models. In commercial APIs, users are generally limited to prompting generalist models, as fine-tuning is costly for the model provider. But as we show, generalist models are neither sufficiently good nor best possible for many practical tasks. Specializing models to concrete tasks of interests, even with relatively small base models and few labeled examples, can provide a simple, practical, and far more accurate solution.

## 1.2 RELATED WORK

**Benchmarks for legal tasks.** LegalBench (Guha et al., 2023) is a recent multi-task benchmark for natural language understanding in legal domains. As of writing, LegalBench consists of 162 tasks gathered from 40 contributors. LegalBench draws on numerous earlier benchmarking efforts in different legal domains, specifically, inference on contracts (Koreeda & Manning, 2021; Hendrycks et al., 2021), merger agreement understanding (Wang et al., 2023), identifying the legal holding of a case (Zheng et al., 2021), statutory reasoning (Holzenberger & Van Durme, 2021), privacy compliance and policy (Wilson et al., 2016; Zimmeck et al., 2019; Ravichander et al., 2019), and identifying unfair clauses in terms of service (Lippi et al., 2019). Bhambhoria et al. (2024) evaluate the performance of general-purpose models on legal question-answering tasks and advocate for the development of open-source models tailored to the legal domain. We extend and strengthen these valuable efforts to benchmark large language models in legal settings. We focus on core legal classification tasks based on the U.S. Supreme Court Database (Spaeth et al., 2023) and the U.S. Courts of Appeals database (Songer). Our evaluation suite measures the performance of models in annotating court opinions, focusing on tasks that are of interest to the field of empirical legal studies. The tasks we study are complementary to those in LegalBench. We do not evaluate our model on LegalBench, since our model is specialized to the Supreme Court and Appeals Court data.

**Large language models for the legal domain.** General-purpose language models are likely to be trained on a substantial amount of legal data because much of this data is publicly available on the internet. For example, the FreeLaw dataset includes a large collection of court opinions (Gao et al., 2021). Legal-BERT (Chalkidis et al., 2020) is a BERT-like transformer model that was pretrained on a few hundred thousand legal documents. The more recent SaulLM models (Colombo et al., 2024b;a) adapt the open-weights Mistral (Jiang et al., 2023; 2024) models to the legal domain both by continual pretraining and instruction-tuning on legal text. In contrast to Lawma, we consider SaulLM to be a general-purpose model for the legal domain, not tailored to any specific legal task. Our approach differs significantly; we focus on developing models specialized for annotation tasks of practical interest to empirical legal studies. We demonstrate that specialization is highly effective, with our Lawma models significantly outperforming all other evaluated LLMs. For a discussion on the adoption of large language models in the legal community, refer to Appendix A.

```
What follows is an opinion from the Supreme Court of the United States.
Your task is to identify  whether the opinion effectively says that the
decision in this case  "overruled" one or more of the Court\'s own
precedents. Alteration also extends to language in the majority opinion
that states that a precedent of the Supreme Court  has been "disapproved,"
or is "no longer good law". Note, however, that alteration does not
apply to cases in which the Court "distinguishes" a precedent.

[COURT OPINION]

Question: Did the the decision of the court overrule one or more of the
Court's own precedents?
A. Yes
B. No

Think step by step. At the end, respond with "The final answer is
[final_answer]",  where [final_answer] is either a single uppercase
letter (A-Z) or a numerical value (e.g., 9, 121).
```

Figure 3: Example task corresponding to the Supreme Court "precedent alteration" variable.

**Data annotation and labeling.** Hall & Wright (2008) provide an overview of the use of human annotators in empirical legal studies. Student coders have been deployed to extract a wide variety of features from legal data. Although student researchers are much less expensive than private attorneys, the costs can quickly become prohibitive. Depending on the size of the document and the complexity of the task, research assistants can label roughly dozens of examples per hour. Projects involving the labeling of hundreds of documents are financially feasible for many legal scholars, but projects involving many thousands of documents are largely impractical. In an example of a larger annotation effort, Frankenreiter et al. (2021) employed human coders to annotate several thousands of corporate charters. Using ChatGPT for a similar task, Frankenreiter & Talley (2024) estimated that employing human coders would have been approximately ten times more costly.

Data annotation and labeling also play a major role in machine learning benchmarks and applications, see, e.g., Aroyo & Welty (2015); Gray & Suri (2019); Hardt & Recht (2022) for background. Dorner & Hardt (2024) give an extended discussion about label quality and annotator disagreement in the context of machine learning benchmarks.

## 1.3 LIMITATIONS

While our fine-tuned models substantially outperform commercial models, we emphasize that our fine-tuned models are still far from perfect, and the variance in accuracy across tasks remains high. Although our work meets the ethical and technical recommendations by Kapoor et al. (2024) for "developers of legal AI", we maintain caution about the use of large language models for consequential legal tasks. To which extent these models are suitable for use in specific applications requires additional substantive investigation. We add that the legal documents we consider are exclusively from either the U.S. Supreme Court or appellate courts in the United States. We cannot speak to how these results may change for tasks in other legal domains within the United States or legal systems in other countries.

## 2 CASELAWQA

In this work, we focus on legal classification tasks. Legal classification tasks range in complexity, from extremely simple tasks that require little specialized knowledge, to highly sophisticated tasks that involve specific legal knowledge, familiarity with legal principles or discourse, and the ability to engage in nuanced analogical or conceptual reasoning. For example, labeling the ideological valence of a decision requires the annotator to understand how specific legal issues map onto contemporary political debates. Labeling the standard of review applied by an appellate court requires detailed knowledge of these standards as well as the ability to parse procedural history. Many legal doctrines are quite complicated, involving multipart tests, nuanced exceptions, and balancing inquiries.

Our reasons to study legal classification tasks are both technical and substantive. From a technical machine learning perspective, these tasks provide highly non-trivial classification problems where even the best models leave much room for improvement. From a substantive legal perspective, efficient solutions to such classification problems have rich and important applications in legal research, see Appendix A.1 for a detailed discussion.

## 2.1 DATA SOURCES

Central to our study are the U.S. Supreme Court Database (Spaeth et al., 2023) (SCDB) and the U.S. Courts of Appeals database (Songer) (USCAD). The SCDB compiles comprehensive information on U.S. Supreme Court decisions from 1946 onward, and includes variables such as case outcomes, issue areas, legal provisions, and vote counts. The USCAD contains detailed information about decisions made by the U.S. Courts of Appeals from 1925 to 1988. It includes data on judicial decisions, panel compositions, and case characteristics. Both databases provide essential tools for scholars conducting quantitative analyses of the judicial system, decision-making, ideological trends, and the impact of various factors on case outcomes.

The SCDB and USCAD have been instrumental in advancing research on judicial decision making within the fields of political science and empirical legal studies (Epstein et al., 2013; Segal & Spaeth, 2002; Martin & Quinn, 2002). These datasets have been used to drive a substantial research program by allowing scholars to systematically analyze large numbers of court cases, uncovering patterns, trends, and factors influencing judicial outcomes. By providing detailed information on case characteristics, judge attributes, and decision outcomes, these databases have enabled researchers to test theories of judicial behavior, examine the impact of ideology on court decisions, and explore the dynamics of judicial decision-making at different levels of the court system. The insights gained from research using these databases have had significant implications for legal practitioners, policymakers, and the broader legal community, contributing to a better understanding of how courts operate and how legal outcomes are shaped.

## 2.2 CONSTRUCTION OF THE CLASSIFICATION TASKS

We use the variables of the USDB and the USCAD to construct a set of classification tasks. We construct a total of 260 distinct classification tasks, 38 of them corresponding to the Supreme Court database and 232 to the U.S. Court of Appeals. The annotations in the USDB and USCAD serve as labels for these classification tasks. For each task, we additionally construct a prompt template consisting of a general description of the task, followed by a multiple choice question containing each of the possible variable codes. We formulate the task description, question, and answer choices by closely following the databases' variable descriptions. See Figure 3 for an example task.

For every case contained in the USDB and the USCAD, we use the provided case citations to search for its corresponding majority opinion of the court on the Caselaw Access Project, a database of digitized court opinions. We match a total of 24,916 court cases, which we divide into a 70%/10%/20% train/validation/test split. That is, models may not train on any of the court cases used for evaluation.

Since many of the classification tasks contain heavily imbalanced classes, we subsample the majority class such that there are at most as many task examples in the majority class as task examples in all other classes combined. As a result, a constant classifier that outputs the majority class label will never achieve more than $50\%$ accuracy on any individual task. This results in a more honest measure of model performance, as models cannot attain high accuracy simply because a task is heavily imbalanced. We report in Appendix E results without subsampling of the majority class.

We plot some statistics of the tasks in Figure 4. First, court opinions tend to be long, with 12% having above 8,000 tokens, the typical maximum context size for current state-of-the-art models, such as Llama 3. Second, some tasks have a large number of classes, with 28% of tasks having more than 10 classes. Third, there is a large variability in terms of the number of task examples, ranging from a couple dozen to 18500 task examples. Our final dataset comprises 718,971 task examples.

To reduce the compute required for evaluating the benchmark, we select at random 5,000 examples from the Supreme Court tasks and 5,000 examples from the Court of Appeals tasks. We include only court cases where the court opinion, including the head matter, contains at least 2,000 characters, ensuring the opinion is at least a few sentences long. These 10,000 task examples comprise the test

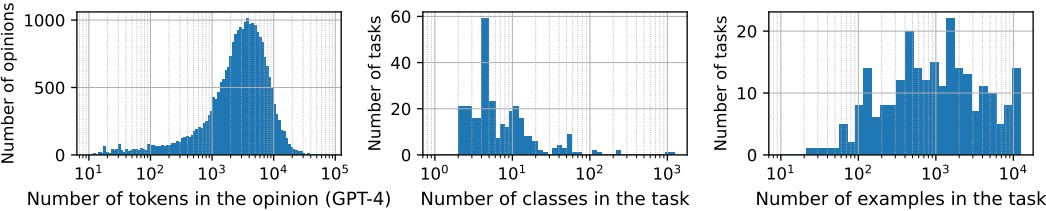

Figure 4: General statistics of the court opinions and legal classification tasks considered.

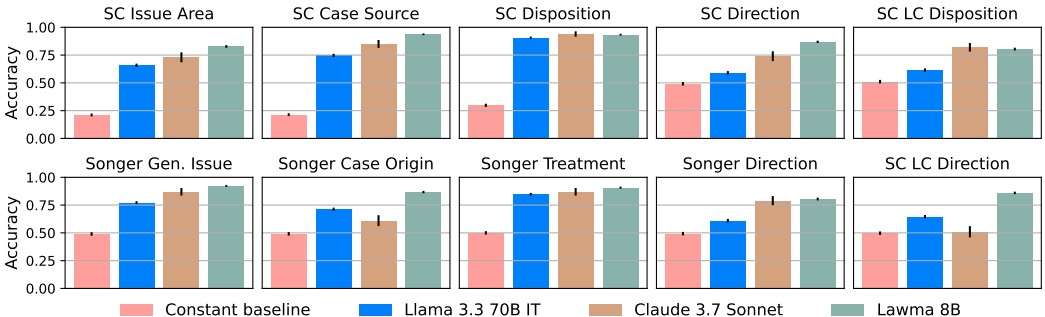

Figure 5: Accuracy of models on ten different legal classification tasks of particular interest.

set of `CaselawQA`. We nonetheless make available all 143,635 task examples corresponding to the test court cases, which we call the *extended test set*. Evaluating on the extended test set is 14x as expensive, but provides much more fine-grained information on models' performance across all 260 legal classification tasks, rather than simply an aggregate measure of model performance. In this work, we report accuracy on the extended test set, unless otherwise stated.

### 2.3 EVALUATION METHODOLOGY

We evaluate models using a prompt template identical to the one for the MMLU benchmark (Hendrycks et al., 2020). Since many popular benchmarks are phrased as multiple-choice questions, recent models tend to do well for them (Dominguez-Olmedo et al., 2024). Due to diverse set of models and large number of tasks under consideration, we perform no prompt tuning.

We use accuracy as the evaluation metric. Given that the tasks we consider involve vastly differing numbers of answer choices, accuracy provides an interpretable measure of performance. Additionally, accuracy is the standard metric used in knowledge-testing LLM benchmarks. For completeness, we also report balanced accuracy and macro-averaged F1 scores in Appendix E.

When reporting aggregate performance across multiple tasks (e.g., all Supreme Court tasks), we compute the average accuracy across all task examples. Intuitively, we can visualize the Supreme Court database as a large table with dimensions corresponding to the number of court cases (rows) and the number of tasks (columns). The aggregate accuracy, in this case, represents the fraction of entries in this table that the model correctly predicts. For completeness, we also report mean task accuracy (i.e., macro-averaging rather than micro-averaging across tasks) in Appendix E.

### 3 EVALUATION BASELINES

We evaluate the performance of various large language models on CaselawQA, our legal annotation benchmark. Among models with open weights, we select for evaluation the prominent Llama 3 (MetaAI, 2024) and Qwen 2.5 (Yang et al., 2024) instruct model families, and the recently released DeepSeek R1 (Guo et al., 2025) reasoning model. Among commercial models, we evaluate GPT-4o 2024-08-06 (Hurst et al., 2024), o3-mini 2025-01-31 (OpenAI, 2025b) (high reasoning effort), GPT-4.5 Preview 2025-02-27 (OpenAI, 2025a) and Claude 3.7 Sonnet (Anthropic, 2025). We also report the performance of the constant classifier that always predicts the majority class for each

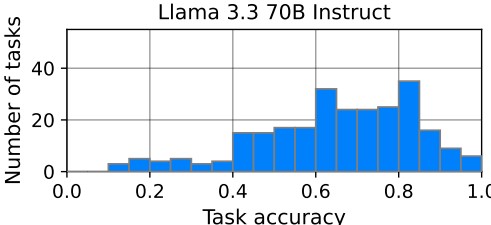 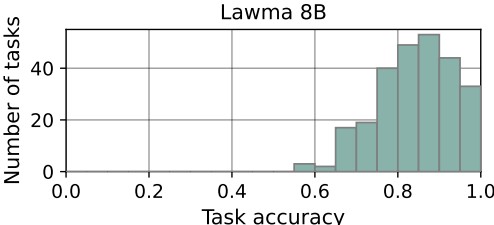

Figure 6: Distribution of task performance for Llama 3.3 70B Instruct and Lawma 8B.

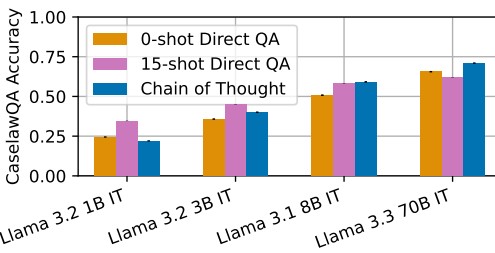 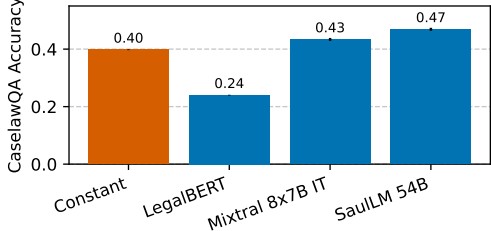

Figure 7: For large models, Chain of Thought prompting outperforms few-shot QA prompting.

Figure 8: General-purpose models for the legal domain perform poorly on CaselawQA.

task, regardless of the Court opinion being labeled. This simple classifier serves as a baseline for non-trivial performance and achieves an accuracy of 40%.

Figure 1 shows the performance of the largest evaluated models, each with at least 70B total parameters. Their accuracy ranges from 70% to 78%, with Claude 3.7 Sonnet achieving the highest performance. All large models outperform the constant classifier baseline by a wide margin. We highlight in Figure 5 the performance of Llama 3.3 70B Instruct and Claude 3.7 Sonnet on ten different tasks of particular importance to empirical legal research. See Appendix B for a description of these highlighted tasks. We observe that performance can be modest even in relatively simple tasks (e.g., < 75% accuracy for the SC Issue Area task). For more complex tasks, Claude 3.7 Sonnet may perform no better than the constant classifier baseline (e.g., for SC LC Direction, that is, identifying the ideological direction of the lower Court's decision).

More broadly, we observe large variance in models' performance across tasks. We plot in Figure 6 left the distribution of tasks' accuracies for Llama 3.3 70B Instruct. While its micro-average accuracy is 71%, it exhibits a reasonably long tail of tasks for which performance is very poor. In fact, for 88 of the tasks (34% of all tasks), Llama 3.3 70B Instruct does not perform significantly better than the trivial constant classifier.

Our evaluations indicate that, while large models generally exhibit non-trivial legal annotation performance, their performance across tasks is highly varied and can be modest even for simple tasks.

**Few-shot and Chain of Thought prompting.** The predominant alternative to CoT prompting is MMLU-style direct question answering ("Direct QA") prompting, where the model is expected to directly output an answer label (e.g., "A" or "B") without first producing a reasoning chain. One benefit of Direct QA is that it is straightforward to include examples in-context. In contrast, few-shot CoT prompting requires collecting reasoning traces for each of the in-context examples.

We compare in Figure 7 the following prompting strategies: zero-shot direct QA, few-shot direct QA, and zero-shot CoT. We consider the Llama 3 Instruct family of models. For the Llama 3 models, we can typically fit 15 examples in-context, since their maximum context window is 128k tokens and each task example is at most 8k tokens. We observe that for the smaller models (i.e., <3B parameters), few-shot Direct QA performs best. In contrast, for the larger models (i.e., >8B parameters), Chain of Thought is superior. In fact, the largest model evaluated few-shot, Llama 3.3 70B Instruct, does not benefit from including examples in-context.

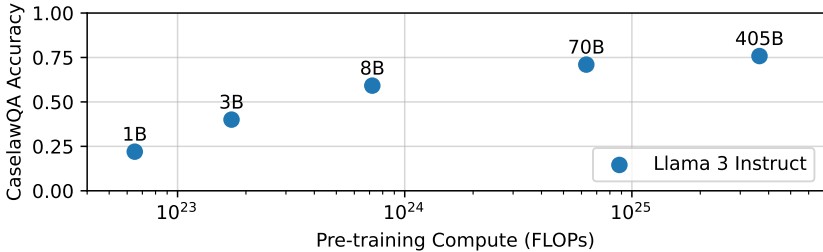

Figure 9: CaselawQA performance as a function of pre-training compute for Llama 3 models of varying scales. Performance improves monotonically with compute, with diminishing returns.

Our results indicate that few-shot prompting is not a fruitful strategy to adapt large models to the legal classification tasks at hand. Since Court opinions tend to be rather lengthy, few examples may fit in-context, potentially preventing the model from improving over the zero-shot baseline.

**Language models for the legal domain.** We additionally evaluate two prominent LLMs adapted to the legal domain: LegalBERT (Chalkidis et al., 2020), a small BERT-style model pre-trained on legal documents, and SaulLM 54B (Colombo et al., 2024a), a Mixtral 7x8B (Jiang et al., 2024) model adapted to the legal domain both by continual pretraining and instruction-tuning on legal text.

We report their CaselawQA performance in Figure 8. We observe that LegalBERT performs poorly, substantially underperforming the constant classifier baseline. This is unsurprising, as LegalBERT is a very small model by today's standards, with only 110M parameters and a context window of 512 tokens, which most of our Court opinions exceed. Regarding SaulLM 54B, we find that it improves upon its base model–Mixtral 8x7B Instruct– by 4 accuracy points. Nonetheless, its legal annotation performance is poor, and lags that of smaller, generalist models such as Llama 3.1 8B Instruct.

**The efficacy of scaling generalist models.** Downstream benchmark performance tends to increase with pre-training compute (Wei et al., 2022; MetaAI, 2024; Gadre et al., 2024; Dominguez-Olmedo et al., 2024). We plot in Figure 9 the performance of the Llama 3 Instruct family of models against their pre-training compute. Similarly to Kaplan et al. (2020), we approximate pre-training compute $C$ in FLOPs as $C \approx 6 \cdot N \cdot D$, where $N$ is model size and $D$ is the number of tokens.

We find that accuracy improves monotonically with pre-training compute. However, we observe signs of diminishing returns. Therefore, further scaling pre-training compute will likely only yield moderate improvements in performance, with great financial cost (e.g., the cost of training LLama 3 405B is in the order of tens of millions of U.S. dollars). Further evidence of the limitations of continuing to scale generalist models is that state-of-the-art commercial models such as GPT-4.5 and Claude 3.7 Sonnet show minimal improvements over Llama 3 405B Instruct, see Figure 1.

## 4  FINE-TUNING AND THE POWER OF SPECIALIZATION

In this section, we present a detailed analysis of how models can be specialized for legal classification tasks. We start by fine-tuning five different models, ranging in size from 135M parameters to 8B parameters, on all 260 legal annotation tasks simultaneously, resulting in our Lawma family of models. We then perform additional fine-tuning experiments highlighting different aspects, its sample efficiency, its generalization to unseen tasks and Courts, and the effect of single task specialization.

### 4.1  THE LAWMA MODELS

We first fine-tune on *all tasks* simultaneously. We fine-tune the following models: HuggingFace's SmolLM2 (Allal et al., 2025) 135M and 360M Instruct, Llama 3.2 (MetaAI, 2024) 1B and 3B Instruct, and Llama 3.1 8B Instruct. We refer to these models as Lawma 135M, Lawma 360M, Lawma 1B, Lawma 3B, and Lawma 8B, respectively. We fine-tune on the 260 classification tasks comprising CaselawQA. The fine-tuning dataset contains a total of 1.96B tokens. We fine-tune for 3

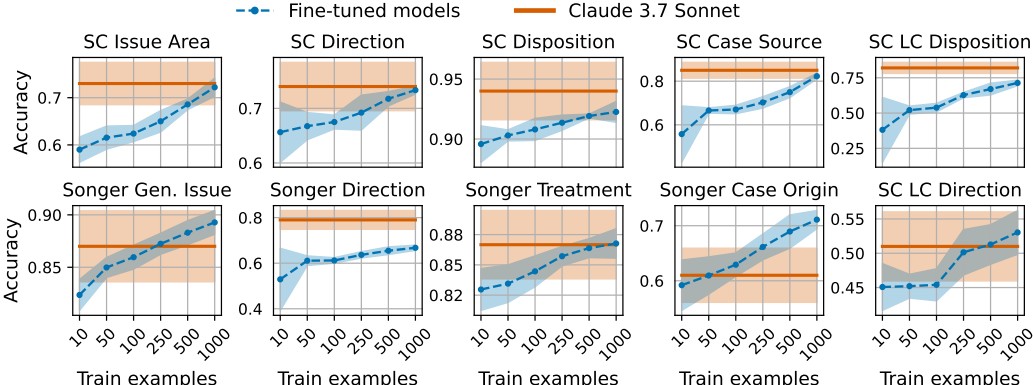

Figure 10: Sample efficiency of fine-tuning Llama 3.1 8B Instruct on a single task. Hundreds of task examples are typically enough to match the zero-shot performance of Claude 3.7 Sonnet. Dashed blue line indicates the accuracy of the fine-tuned model as a function of the number of training examples. The shaded area indicates the 95% confidence interval over the randomly sampled training examples (5 random seeds)

.

epochs. We find that additional epochs do not significantly improve performance. See Appendix F for additional details regarding the model training.

We compare in Figure 2 the performance of the Lawma models with the largest Llama 3 model, Llama 3 405B, and the best-performing commercial model, Claude 3.7 Sonnet. We observe that performance of the specialized models improves with model size. Remarkably, the smallest specialized model, Lawma 135M, is competitive with both Llama 3 405B and Claude 3.7 Sonnet, while being several orders of magnitude smaller. In addition, the largest Lawma model, Lawma 8B, substantially outperforms Claude 3.7 Sonnet, achieving 87% accuracy on CaselawQA. Moreover, for the 10 legal annotation tasks highlighted in Figure 5, Lawma 8B outperforms Claude 3.7 Sonnet in 7 of them, and matches its performance on the remaining 3. For some of the tasks, the performance improvements are very large. For example, Lawma 8B outperforms Claude 3.7 Sonnet by 30 accuracy points on SC LC Direction and 20 accuracy points on Songer Case Origin.

We plot the distribution of task accuracies of Lawma 8B in Figure 6 right. While generalist models can exhibit a long tail of tasks with very poor performance, Lawma 8B achieves not only higher performance but also smaller variance in its task accuracies, which generally lie in the 75% to 95% accuracy range. Nonetheless, the variance in accuracy across tasks remains reasonably high, and the model performs poorly for a substantial number of tasks.

## 4.2 Sample efficiency

We study how task accuracy scales as models fine-tune on more training examples. We consider the 10 tasks highlighted in Section B. We fine-tune Llama 3.1 8B Instruct on each task independently, rather than on all tasks simultaneously as in the previous experiments. For each task, we fine-tune on 10, 50, 100, 250, 500, and 1000 task examples. We select task examples uniformly at random, and train 5 different models corresponding to different random seeds on the examples selected for training. We therefore fine-tune and evaluate a total of $10 \cdot 6 \cdot 5 = 300$ models. We fine-tune for a maximum of 20 epochs and early stop when validation loss increases for 3 consecutive epochs.

Figure 10 shows how accuracy improves with the number of training examples. We additionally plot the accuracy of Claude 3.7 Sonnet, the best-performing commercial model. Due to the high cost of evaluating Claude 3.7 Sonnet, we only evaluate 100 examples per task. We observe that hundreds of training examples are enough to match or beat the Claude 3.7 Sonnet baseline for 7 out of the 10 highlighted tasks. This is crucial, since labeling a few hundred data points is often financially feasible for many legal scholars (Hall & Wright, 2008). With relative few labelled task

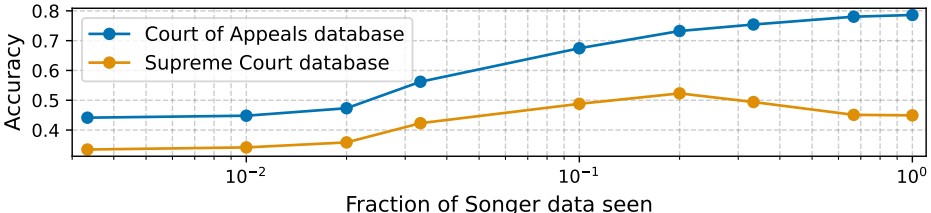

Figure 11: Training on the Court of Appeals tasks improves accuracy on Supreme Court tasks.

examples, fine-tuning reasonably small publicly available models can be competitive with state-of-the-art closed models. Moreover, accuracy continues to improve with additional examples.

### 4.3 GENERALIZATION TO UNSEEN DATABASES

We now investigate whether fine-tuning only on the Songer Appeals Court database allows us to generalize to the Supreme Court database. We fine-tune Llama 3 8B Inst for one epoch on all Songer tasks simultaneously. We plot in Figure 11 the mean accuracy for Court of Appeals tasks and Supreme Court tasks at intermediate checkpoints. As expected, performance on Court of Appeals tasks improves monotonically with the number of training examples seen. More interestingly, we observe that mean task accuracy for the Supreme Court also improves substantially, by up to 18.8 accuracy points at 20% of the training steps. Thereafter, performance degrades, seemingly plateauing at 11.3 accuracy points above the non-finetuned performance of Llama 3 8B Inst.

Our findings indicate that, since there is some degree of overlap between Court of Appeal and Supreme Court tasks, fine-tuning on the former transfers to the latter. This suggests that Lawma might be of practical use beyond the Supreme Court and Court of Appeals tasks it was trained on.

Note, however, that fine-tuning only on the Court of Appeals database results in a mean case accuracy of 51.6%, compared to 82.4% for Lawma 8B. That is, not fine-tuning on Supreme Court cases results in a 30.9 accuracy points decrease in performance. These results again highlight the importance of fine-tuning precisely on the target tasks of interest.

## 5 DISCUSSION

The cost of human annotators represents a considerable bottleneck for the field of empirical legal studies. In many scientific disciplines, the advent of low-cost and flexible tools for data extraction can lead to tremendous boosts in scholarly productivity and knowledge production. For example, the falling cost of genetic sequencing led to a paradigm shift across the biological sciences, as genetic data became increasingly available in fields as disparate as public health and entomology (Köser et al., 2012; Ballare et al., 2019). A flexible automated feature extraction tool for legal texts holds similar potential for empirical legal studies, as a large realm of conceivable but impractically expensive research projects becomes accessible.

The generalist abilities of large language models are vital for commercial APIs, where users are largely restricted to prompting. But as we show, generalist models are neither sufficiently good nor best possible for classification tasks that arise in empirical legal work. Lightly fine-tuned special purpose models achieve significantly higher accuracy from relatively few labeled examples. Labeling a few hundred cases is often financially feasible. This suggests a simple and practical strategy for solving legal classification tasks: Obtain a few hundred labeled examples, fine-tune an open weights model, and use the fine-tuned model to annotate the remaining cases.

The tasks we introduce are also interesting from a benchmarking perspective. The accuracy numbers are neither too low nor too high. The best models achieve non-trivial, but modest zero-shot performance. And even fine-tuned models don't reach intercoder agreement rates. This situation suggests that these legal classification tasks may be good test cases for future model advances. As such, we hope to extend and strengthen existing evaluation efforts.

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

## A ADOPTION OF LARGE LANGUAGE MODELS IN THE LEGAL COMMUNITY.

The legal community has moved relatively quickly in adopting GPT models. Several startups have begun using incorporating large language models, including GPT, into legal products (Wiggers, 2022). Lexis Nexis, a major commercial provider of law-related services, has partnered with Open AI and Anthropic to offer legal text generation (LexisNexis, 2023). Legal scholars have evaluated GPT's performance on the bar exam (Katz et al., 2024) as well as law school exam (Choi et al., 2023). Choi & Schwarcz (2023) examined how GPT-4 can improve student performance on law school exams. Nay et al. (2024) examined how LLMs perform on answering multiple choice questions related to tax law. Gray et al. (2024) used GPT models to extract information from cases concerning the factors that predict the constitutionality of police stops. Choi (2023) used GPT-4 to extract information concerning interpretative techniques from U.S. Supreme Court decisions. Livermore et al. (2023) tested the performance of GPT models for categorizing cases by issue areas and in recommending citations based on case similarity. Savelka & Ashley (2023) evaluate the zero-shot performance of GPT-4 on a variety of semantic legal annotation tasks. Engel & Mcadams (2024) ask GPT for the ordinary meaning of statutory terms. In the area of corporate law, Frankenreiter & Talley (2024) use GPT-4 to extract information about the contents of corporate charters.

### A.1 POTENTIAL APPLICATIONS OF EFFICIENT SOLUTIONS TO LEGAL CLASSIFICATION TASKS

More efficient ways to solve legal classification tasks would be tremendously useful in practice. A well functioning system to automatically extract relevant features from legal texts could, in particular, facilitate empirical legal study across a wide range of domains. This research could include not only social scientific study of the causes or consequences of judicial decisions, but also more traditional research modalities based on doctrinal interpretation (Livermore & Rockmore, 2019). There is an almost unlimited variety of features that legal scholars could study, ranging from the factors cited by judges when deciding the outcomes of property law disputes to the relationship between the party affiliation of judges and their use of different interpretative styles. With the digitization of legal texts at the U.S. state level and outside the U.S., low-cost and flexible featurization can also boost efforts to show the geographic diffusion of legal concepts.

## B    Highlighted tasks

Throughout this paper, as in Figure 5, we provide detailed results for ten tasks. Six of these tasks are from the SCDB, and four are from the USCAD. We selected tasks that we believe are particularly relevant to the legal community and chose tasks with varying levels of complexity, ranging from relatively simple (e.g., determining the issue area) to more complex (e.g., determining the ideological 'direction' of the court decision).

Four tasks from the USCAD and all tasks from the SCDB were selected to form pairs, with each pair consisting of one task from the USCAD and one from the SCDB that capture similar concepts. It is important to note that, despite capturing broadly similar concepts, the precise formulation of the tasks might differ between the USCAD and the SCDB, making them less than perfectly comparable. In addition to the four pairs, we include two tasks from the SCDB that involve determining features of the decision reviewed by the Supreme Court on the basis of the Supreme Court opinion. The following is a description of the task pairs:

- **SC Issue Area / Songer Gen Issue:** These tasks capture the case's issue area, requiring a determination of whether the case belongs to one of several broadly defined categories, such as criminal cases or First Amendment cases. These tasks are expected to be of relatively low complexity.
- **SC Case Source / Songer Case Origin:** These tasks require identifying the court or adjudication body where the case was originally initiated before moving up the judicial hierarchy. Like the previous pair, these tasks are expected to be of relatively low complexity.
- **SC Disposition / Songer Treatment:** These tasks involve determining how the deciding court treated the lower court opinion it reviewed, such as whether it affirmed or reversed the opinion. We consider these tasks to be of relatively low complexity.
- **SC Direction / Songer Direction:** These tasks involve determining the ideological 'direction' of the decision, specifically whether the decision supports a "conservative" or "liberal" outcome. We consider these tasks to be comparably complex.
- **SC LC Disposition / SC LC Direction:** These tasks involve determining the disposition and ideological 'direction' of the decision reviewed by the Supreme Court. As these tasks require analyzing features of another decision based on the text of the Supreme Court decision, we consider these tasks to be comparably complex.

## C    Intercoder agreement analysis

The Songer Appeals Court database provides intercoder agreement rates for a subset of the variables. These intercoder agreement rates provide valuable context for the performance of our model. Specifically, intercoder agreement gives us information about the inherent label noise in the annotation procedure. In particular, the intercoder agreement rate gives a natural upper bound on model performance, as we cannot expect the model to perform well when the label is uncertain or subject to interpretation.

However, we cannot directly compare intercoder agreement rates with the accuracy numbers we report. The reason is that in each task we subsampled the majority class to be no larger than the union of all other classes. This is a design choice we made to account for class imbalance. In this section, we map our model's accuracy to *adjusted* accuracy numbers that undo the subsampling step. This results in accuracy numbers that are commensurate with the intercoder agreement rate.

Table 1 considers several tasks from the Appeals Court database, including the selected ones we highlighted in various figures. Each row corresponds to one task and provides the intercoder agreement rate, adjusted (and unadjusted) accuracy achieved by Lawma 8B, and the fraction of samples we retained in the majority class. A fraction of 100% means that we kept all samples. The smaller the fraction the larger the majority class is relative to the other classes.

The table contains several interesting insights:

- The adjusted accuracy of Lawma 8B is generally within single digit percentage points of the intercoder agreement rate for easy tasks such as general issue classification (GENISS).

| Name | IC Agreement | Adj accuracy | (unadjusted) | Keep |
|---|---|---|---|---|
| WEIGHTEV (songer_weightev) | 76 | 78.7% | (77.2%) | 28.72% |
| PROCEDUR (songer_procedur) | 78 | 75.2% | (73.9%) | 83.08% |
| ORIGIN (songer_origin) | 83.2 | 80.1% | (77.7%) | 53.13% |
| DIRECT2 (songer_direct2) | 85.6 | 67.5% | (67.5%) | 100.00% |
| DIRECT1 (songer_direct1) | 94 | 80.5% | (80.5%) | 100.00% |
| TREAT (songer_treat) | 95.2 | 91.1% | (90.1%) | 71.26% |
| GENISS (songer_geniss) | 97.6 | 93.2% | (92.9%) | 84.77% |
| CIRCUIT (songer_circuit) | 100 | 93.2% | (93.2%) | 100.00% |
| COMMENT (songer_comment) | 100 | 100.0% | (91.7%) | 0.13% |

Table 1: Intercoder agreement rates, Lawma accuracies, and fraction of the majority class retained in our sample. Rows are sorted in increasing order of agreement rate.

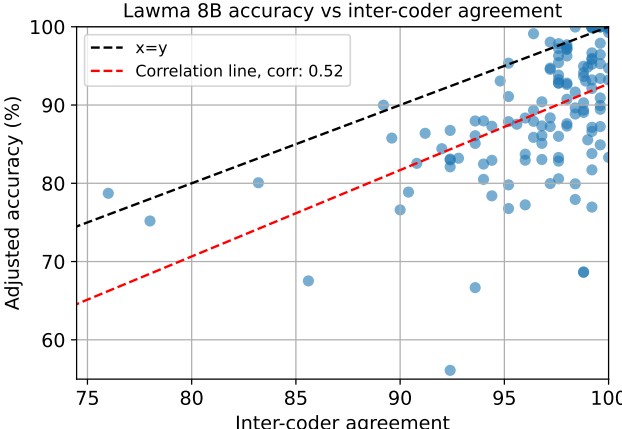

Figure 12: Lawma task accuracy against inter-coder agreement. Lawma

- Lawma 8B is surprisingly close on the two tasks with the lowest intercoder reliability, i.e., WEIGHTEV and PROCEDUR. This shows that high intercoder reliability is no prerequesite for the model to perform well, i.e., close to the agreement rate.

- On harder tasks, like identifying the ideological valence of a decision (DIRECT1 and DIRECT2), Lawma 8B is below the agreement rate by double digit percentage points.

- Tasks with very high agreement rate (e.g., CIRCUIT and COMMENT) are not all alike. Some of them (e.g., COMMENT) correspond to a task with extreme class imbalance. Here, the model reaches the agreement rate. Other tasks (e.g., CIRCUIT) have perfect agreement rate, no class imbalance, and yet Lawma is far from the agreement rate.

These findings speak to the task heterogeneity and the non-trivial nature of the task suite as a classification benchmark.

## D  SPECIALIZING FOR SINGLE TASKS

We now study how much accuracy we stand to gain by fine-tuning on a *single* task. We specialize models for each of the 10 tasks highlighted in Section B. We specialize the follow models: Llama 3 8B Inst, Llama 3 8B Inst fine-tuned for one epoch on all tasks, and Lawma 8B (i.e., Llama 3 8B Inst fine-tuned for three epochs on all tasks). For each task, we fine-tune for a maximum of 20 epochs and early stop when validation loss increases for 3 consecutive evaluation stpes, each corresponding to one tenth of an epoch.

Figure 13 shows the results of specialization to single tasks. First, we observe that, for 7 out of 10 tasks, Llama 3 8B Inst fine-tuned on all tasks for one epoch (yellow) outperforms Llama 3 8B

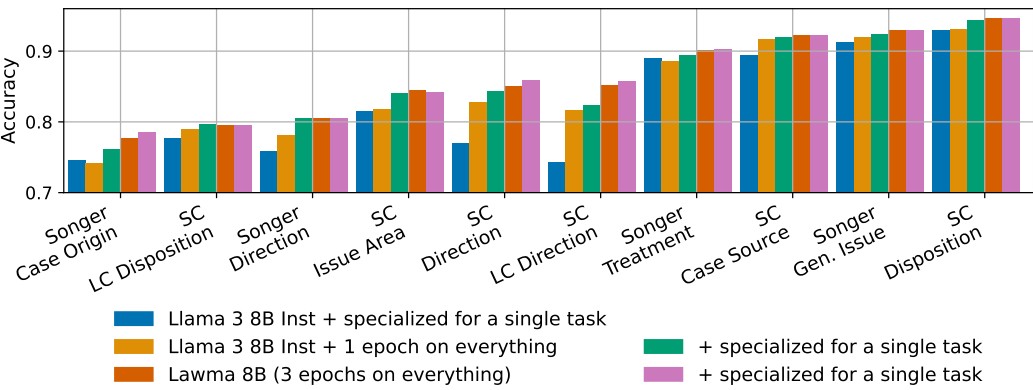

Figure 13: Specializing Lawma 8B to individual tasks can yield small improvements in accuracy.

Inst specialized for a single task (blue). That is, there is value to fine-tuning on our entire dataset rather than overspecializing for a single task. One explanation is that there is substantial cross-task overlap, and fine-tuning on the entire dataset amounts training on many more examples –even if on average these examples are less relevant.

Secondly, we observe that after fine-tuning on *all* 260 tasks for 1 epoch (yellow), further specializing for a single task (green) improves performance on all cases. Importantly, the latter outperforms the specialized Llama 3 8B Inst (blue) in all tasks. That is, a model that is fine-tuned on everything provides a "better" foundation from which to then "overspecializing" for a single task.

Thirdly fine-tuning on everything for three epochs (i.e., Lawma 8B, in red) again improves over the specialized models (i.e., green). Lastly, "overspecializing" Lawma 8B for a single task results in small single digit improvements for 3 out of the 10 tasks. However, we observe no benefits from specializing Lawma 8B for most (7/10) of the tasks.[1] These results show that we don't leave much accuracy on the table by fine-tuning a single model for all tasks. This is practically quite appealing, since it obviates the need to maintain a separate model for each task. A single model suffices.

# E  ADDITIONAL PERFORMANCE RESULTS

## E.1  BALANCED ACCURACY AND MACRO-F1

See Figure 14 and Figure 15 for evaluation results using mean balanced accuracy and mean macro-F1 as the evaluation metric, respectively.

## E.2  RESULTS WITHOUT SUBSAMPLING THE MAJORITY CLASS

Figure 16 presents the evaluation results when not subsampling the majority class. Models achieve very hight accuracy on many tasks simply because they correctly identify the majority class.

## E.3  AVERAGE TASK ACCURACY RESULTS

Figure 17 presents the results when using mean task accuracy across tasks as the evaluation metric.

## E.4  COMPARING LLAMA 70B INSTRUCT AND GPT-4 TO THE CONSTANT CLASSIFIER

Figure 18 illustrates the difference in performance across tasks between GPT-4 and Llama 3 70B Instruct, and the majority class classifier. GPT-4 and Llama 3 70B Instruct perform worse than the constant classifier for dozens of tasks.

---

[1]There is a small decrease in performance for SC Issue Area. This is because early stopping is performed with respect to loss on the validation set, but models are evaluated for accuracy on the test set.

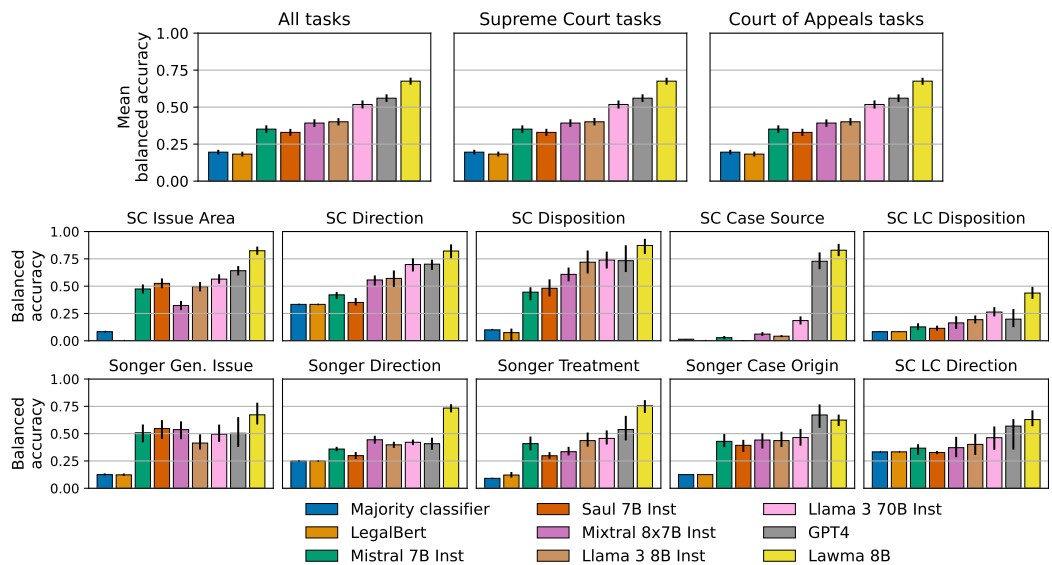

Figure 14: Evaluation results using balanced accuracy as the evaluation metric.

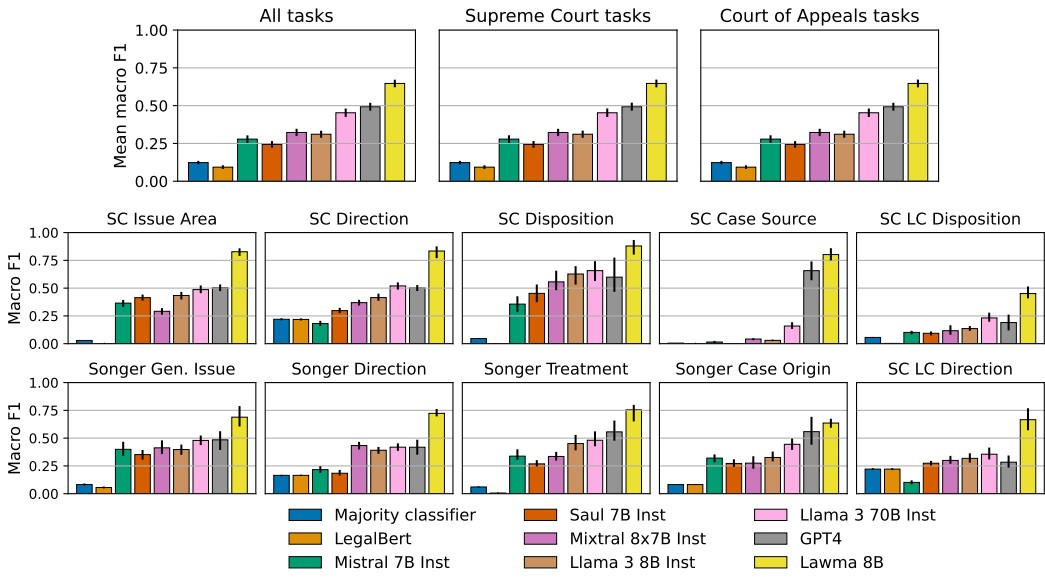

Figure 15: Evaluation results using mean macro-F1 as the evaluation metric.

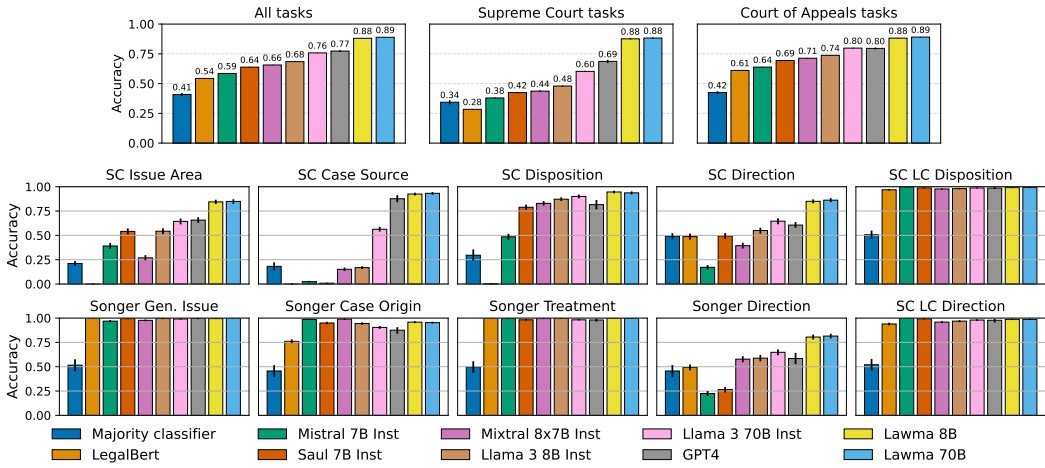

Figure 16: Evaluation results without subsampling the majority class.

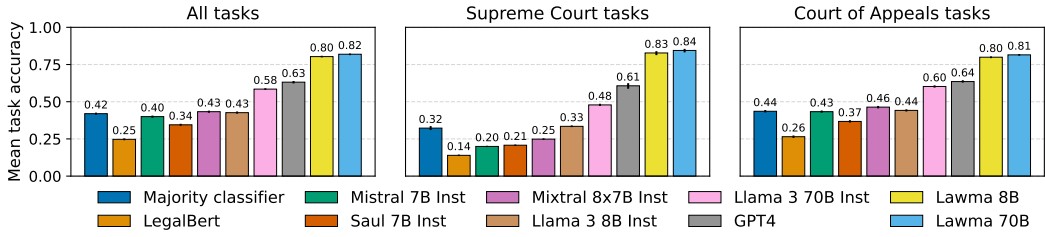

Figure 17: Evaluation results when using mean task accuracy across tasks as the evaluation metric.

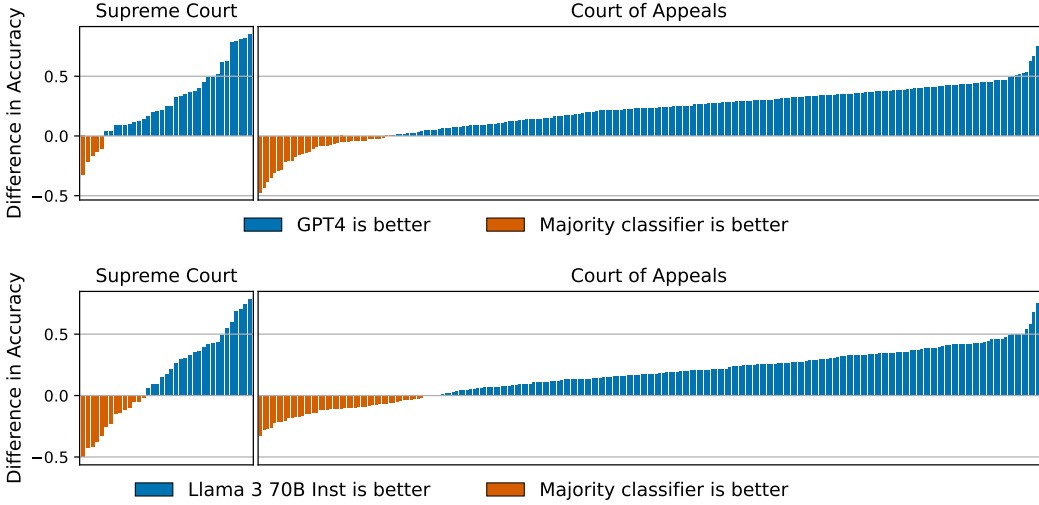

Figure 18: Difference in zero-shot accuracy between GPT4, Llama 3 70B Instruct, and the majority classifier. Each vertical bar represents the accuracy difference on one task, sorted in ascending order.

### E.5 CHAIN OF THOUGHT EVALUATION

We follow the standard methodology of eliciting CoT by appending to the prompt "Let's think step by step." Since CoT requires two orders of magnitude more compute for evaluation than the standard QA approach, we only evaluate Llama 3 8B Instruct and Llama 3 70B Instruct. This required over

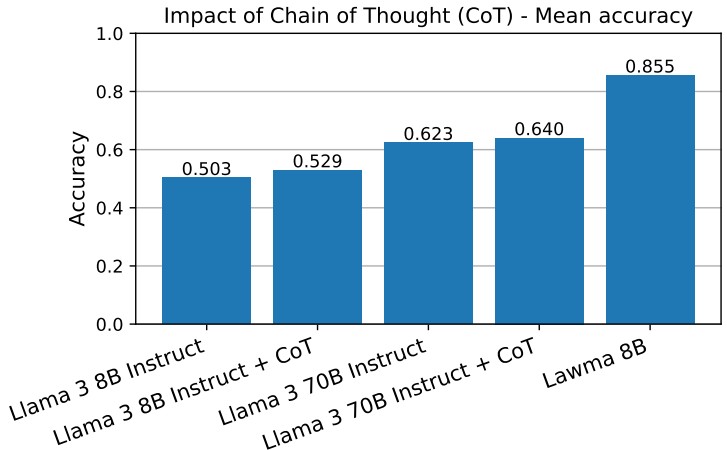

Figure 19: Performance improvements using Chain of Thought (CoT).

500 H100 GPU hours. We observe that CoT leads to modest improvements of performance for both the 8B and 70B model, on average of 2 to 3 accuracy points, see Figure 19. Nonetheless, Lawma 8B still strongly outperforms Llama 3 70B, by over 20 accuracy points.

## F   FINE-TUNING DETAILS

**Compute requirements.**   We fine-tune on a cluster consisting of NVIDIA H100 GPUs. Fine-tuning on all tasks simultaneously required approximately 600 H100 hours for the 8B model and 1600 GPU hours for the 70B model. In total, the experiments presented in the paper required approximately 8000 H100 GPU hours.

### F.1   LAWMA

We fine-tuning with a maximum sequence length of 8192 tokens. We use the AdamW optimizer with full precision, $\beta_1 = 0.9$, $\beta_2 = 0.95$, $\epsilon = 10^{-8}$. We use a peak learning rate of $2 \cdot 10^{-6}$. We use a cosine learning rate schedule, with 180 warm-up steps (approx. 4% of a full epoch) and decay to $10\%$ of the peak learning rate. We use a weight decay of 0.1. We clip gradient to 1.0 max norm. We pack samples using the axolotl library (Cloud, 2024), which improves training efficiency by approximately 40%. For Lawma 8B, we fine-tune Llama 3 8B Instruct for 3 epochs. We train on a node of 7 H100s using DeepSpeed Zero 2, with a global batch size of 56. For Lawma 70B, we fine-tune Llama 3 70B Instruct for 1 epoch. We train on 8 nodes of 8 H100s each using DeepSpeed Zero 3, with a global batch size of 64. We find that additional epochs hurt average task performance, although performance continues to improve for some of the tasks.

### F.2   ADDITIONAL FINE-TUNING EXPERIMENTS

The hyperparameters are identical to those used for Lawma unless otherwise specified.

**Scaling experiments.**   We fine-tune the Pythia and Llama 2 models with a peak learning rate of $2 \cdot 10^{-5}$, which we find to be result in higher performance than a peak learning rate of $2 \cdot 10^{-6}$. For the Llama 3 models, we use a learning rate of $2 \cdot 10^{-6}$, which we find to be perform better than $2 \cdot 10^{-5}$. We fine-tune for a single epoch. We use a batch size 64. We fine-tune models with their pretraining max sequence length, that is, 2k tokens for Pythia, 4k tokens for Llama 2, and 8k tokens for Llama 3. We use a warm up ratio of 0.03. Due to the costs associated with training the 70B model, we simply take Lawma 70B rather than re-training the model with these slightly different training hyperparameters.

**Sample efficiency and specialization**    We fine-tune for up to 20 epochs. We evaluate the loss on a separate validation set and early stop if the loss increases for 3 consecutive evaluation steps. For the sample efficiency experiments, we evaluate at the end of every epoch. For the specialization experiments, we evaluate every 0.1 epochs. We decay the learning rate to 10% of the peak learning rate over the 20 epochs. We fine-tune with a batch size of 64. For the specialization experiments, we train models both with and without learning rate warm up, and report the accuracy of the best model. We use the AdamW BitsAndBytes 8-bit optimizer, allowing us to fine-tune the models in a single H100 GPU.

**Generalization**    We fine-tune only on the Songer Court of Appeals tasks. We fine-tune with batch size 64. We fine-tune for one epoch and we checkpoint models at 10, 30, 60, 100, 300, 600, 1000, 2000, and 3000 training steps. A full epoch on the Songer Court of Appeal tasks corresponds to 3096 training steps.

## G  LIST OF ALL TASKS

| Variable | Question | Sample answer choices |
|---|---|---|
| sc_adminaction | What is the agency involved in the administrative action? | Army and Air Force Exchange Service, Atomic Energy Commission, Secretary or administrative unit or personnel of the U.S. Air Force |
| sc_adminaction_is | Did administrative action occur in the context of the case? | No, Yes |
| sc_adminactionstate | What is the state of the state agency associated with the administrative action? | Alabama, Alaska, American Samoa |
| sc_authoritydecision | What is the basis of the Supreme Court's decision? | judicial review (national level), judicial review (state level), Supreme Court supervision of lower federal or state courts or original jurisdiction |
| sc_casedisposition | What is the disposition of the case, that is, the treatment the Supreme Court accorded the court whose decision it reviewed? | stay, petition, or motion granted, affirmed (includes modified), reversed |
| sc_caseorigin | What is the court in which the case originated? | U.S. Court of Customs and Patent Appeals, U.S. Court of International Trade, U.S. Court of Claims, Court of Federal Claims |
| sc_caseoriginstate | What is the state of the court in which the case originated? | Alabama, Alaska, American Samoa |
| sc_casesource | What is the court whose decision the Supreme Court reviewed? | U.S. Court of Customs and Patent Appeals, U.S. Court of International Trade, U.S. Court of Claims, Court of Federal Claims |
| sc_casesourcestate | What is the state of the court whose decision the Supreme Court reviewed? | Alabama, Alaska, American Samoa |
| sc_certreason | What reason, if any, does the court give for granting the petition for certiorari? | case did not arise on cert or cert not granted, federal court conflict, federal court conflict and to resolve important or significant question |
| sc_decisiondirection | What is the ideological direction of the decision? | Conservative, Liberal, Unspecifiable |
| sc_decisiontype | What type of decision did the court make? | opinion of the court (orally argued), per curiam (no oral argument), decrees |

| | | |
|---|---|---|
| sc_declarationuncon | Did the Court declare unconstitutional an act of Congress; a state or territorial statute, regulation, or constitutional provision; or a municipal or other local ordinance? | No declaration of unconstitutionality, Act of Congress declared unconstitutional, State or territorial law, regulation, or constitutional provision unconstitutional |
| sc_issue_1 | What is the issue of the decision? | subconstitutional fair procedure: fugitive from justice, self-incrimination, immunity from prosecution, cruel and unusual punishment, death penalty (cf. extra legal jury influence, death penalty) |
| sc_issue_10 | What is the issue of the decision? | federal pre-emption of state legislation or regulation. cf. state regulation of business. rarely involves union activity. Does not involve constitutional interpretation unless the Court says it does., federal pre-emption of state legislation or regulation. cf. state regulation of business. rarely involves union activity. Does not involve constitutional interpretation unless the Court says it does., national supremacy: public utilities (cf. federal public utilities regulation) |
| sc_issue_11 | What is the issue of the decision? | non-real property dispute between states, non-real property dispute between states, boundary dispute between states |
| sc_issue_12 | What is the issue of the decision? | federal taxation, typically under provisions of the Internal Revenue Code, federal taxation, typically under provisions of the Internal Revenue Code, federal taxation of gifts, personal, business, or professional expenses |
| sc_issue_2 | What is the issue of the decision? | sex discrimination (excluding sex discrimination in employment), Voting Rights Act of 1965, plus amendments, juveniles (cf. rights of illegitimates) |
| sc_issue_3 | What is the issue of the decision? | libel, privacy: true and false light invasions of privacy, parochiaid: government aid to religious schools, or religious requirements in public schools, First Amendment, miscellaneous (cf. comity: First Amendment) |
| sc_issue_4 | What is the issue of the decision? | due process: takings clause, or other non-constitutional governmental taking of property, due process: miscellaneous (cf. loyalty oath), the residual code, due process: miscellaneous (cf. loyalty oath), the residual code |
| sc_issue_5 | What is the issue of the decision? | Freedom of Information Act and related federal or state statutes or regulations, abortion: including contraceptives, abortion: including contraceptives |
| sc_issue_6 | What is the issue of the decision? | attorneys' and governmental employees' or officials' fees or compensation or licenses, commercial speech, attorneys (cf. commercial speech), attorneys' and governmental employees' or officials' fees or compensation or licenses |

| sc_issue_7 | What is the issue of the decision? | labor-management disputes: right to organize, union-union member dispute (except as pertains to union or closed shop), labor-management disputes: employee discharge |
|---|---|---|
| sc_issue_8 | What is the issue of the decision? | natural resources - environmental protection (cf. national supremacy: natural resources, national supremacy: pollution), Employee Retirement Income Security Act (cf. union trust funds), election of remedies: legal remedies available to injured persons or things |
| sc_issue_9 | What is the issue of the decision? | standing to sue: private or implied cause of action, judicial administration: review of non-final order, judicial administration: jurisdiction or authority of federal district courts or territorial courts |
| sc_issuearea | What is the issue area of the decision? | Criminal Procedure, Civil Rights, First Amendment |
| sc_jurisdiction | What is the manner in which the Court took jurisdiction? | cert, appeal, bail |
| sc_lcdisagreement | Does the court opinion mention that one or more of the members of the court whose decision the Supreme Court reviewed dissented? | Yes, No |
| sc_lcdisposition | What treatment did the court whose decision the Supreme Court reviewed accorded the decision of the court it reviewed? | stay, petition, or motion granted, affirmed, reversed |
| sc_lcdispositiondirection | What is the ideological direction of the decision reviewed by the Supreme Court? | Conservative, Liberal, Unspecifiable |
| sc_partywinning | Consider that the petitioning party lost if the Supreme Court affirmed or dismissed the case, or denied the petition. Consider that the petitioning party won in part or in full if the Supreme Court reversed, reversed and remanded, vacated and remanded, affirmed and reversed in part, affirmed and reversed in part and remanded, or vacated the case. Did the petitioning win the case? | Yes, No |
| sc_petitioner | Who is the petitioner of the case? | attorney general of the United States, or his office, specified state board or department of education, city, town, township, village, or borough government or governmental unit |
| sc_petitionerstate | What state is associated with the petitioner? | Alabama, Alaska, American Samoa |
| sc_precedentalteration | Did the the decision of the court overrule one or more of the Court's own precedents? | Yes, No |
| sc_respondent | Who is the respondent of the case? | attorney general of the United States, or his office, specified state board or department of education, city, town, township, village, or borough government or governmental unit |
| sc_respondentstate | What state is associated with the respondent? | Alabama, Alaska, American Samoa |

| | | |
|---|---|---|
| sc_threejudgefdc | Was the case heard by a three-judge federal district court? | Yes, No |
| songer_abusedis | Did the court conclude that it should defer to agency discretion? For example, if the action was committed to agency discretion. | No, Yes, Mixed answer |
| songer_adminrev | What federal agency's decision was reviewed by the court of appeals? | Benefits Review Board, Civil Aeronautics Board, Civil Service Commission |
| songer_agen_acq | Did the court rule for the government in an issue related to agency acquisition of information (e.g. physical inspections, searches, subpoenas, records, etc)? | No, Yes, Mixed answer |
| songer_alj | Did the court support the decision of an administrative law judge? | No, Yes, Mixed answer |
| songer_altdisp | Did the court's ruling on an issue arising out of an alternative dispute resolution process (ADR, settlement conference, role of mediator or arbitrator, etc.) favor the appellant? | No, Yes, Mixed answer |
| songer_amicus | Was there any amicus participation before the court of appeals? | no amicus participation on either side, 1 separate amicus brief was filed, 2 separate amicus briefs were filed |
| songer_app_stid | What is the state of the first listed state or local government agency that is an appellant? | not, Alabama, Alaska |
| songer_appbus | What is the total number of appellants in the case that fall into the category "private business and its executives"? Answer with a number. | N/A |
| songer_appel1_1_2 | This question concerns the first listed appellant. The nature of this litigant falls into the category "private business (including criminal enterprises)". What is the scope of this business? | local, neither local nor national, national or multi-national |
| songer_appel1_1_3 | This question concerns the first listed appellant. The nature of this litigant falls into the category "private business (including criminal enterprises)". What category of business best describes the area of activity of this litigant which is involved in this case? | agriculture, mining, construction |
| songer_appel1_1_4 | This question concerns the first listed appellant. The nature of this litigant falls into the category "private business (including criminal enterprises)", specifically "agriculture". What subcategory of business best describes this litigant? | single family farm, commercial farm, agri-business, farm - other |

| | | |
|---|---|---|
| songer_appel1_2_2 | This question concerns the first listed appellant. The nature of this litigant falls into the category "private organization or association". What category of private associations best describes this litigant? | business, trade, professional, or union (BTPU), other |
| songer_appel1_2_3 | This question concerns the first listed appellant. The nature of this litigant falls into the category "private organization or association", specifically "business, trade, professional, or union (BTPU)". What subcategory of private association best describes this litigant? | Business or trade association, utilities co-ops, Professional association - other than law or medicine |
| songer_appel1_3_2 | This question concerns the first listed appellant. The nature of this litigant falls into the category "federal government (including DC)". Which category of federal government agencies and activities best describes this litigant? | cabinet level department, courts or legislative, agency whose first word is "federal" |
| songer_appel1_3_3 | This question concerns the first listed appellant. The nature of this litigant falls into the category "federal government (including DC)", specifically "cabinet level department". Which specific federal government agency best describes this litigant? | Department of Agriculture, Department of Commerce, Department of Defense (includes War Department and Navy Department) |
| songer_appel1_4_2 | This question concerns the first listed appellant. The nature of this litigant falls into the category "sub-state government (e.g., county, local, special district)". Which category of substate government best describes this litigant? | legislative, executive/administrative, bureaucracy providing services |
| songer_appel1_4_3 | This question concerns the first listed appellant. The nature of this litigant falls into the category "sub-state government (e.g., county, local, special district)", specifically "legislative". Which specific substate government agency best describes this litigant? | City/county council, School Board, board of trustees for college or junior college, Other legislative body |
| songer_appel1_5_2 | This question concerns the first listed appellant. The nature of this litigant falls into the category "state government (includes territories & commonwealths)". Which category of state government best describes this litigant? | legislative, executive/administrative, bureaucracy providing services |
| songer_appel1_5_3 | This question concerns the first listed appellant. The nature of this litigant falls into the category "state government (includes territories & commonwealths)", specifically "legislative". Which specific state government agency best describes this litigant? | Legislature or separate house as an organization, Legislative Committee or Commission, Other Legislative Unit |

| | | |
|---|---|---|
| songer_appel1_7_2 | This question concerns the first listed appellant. The nature of this litigant falls into the category "natural person (excludes persons named in their official capacity or who appear because of a role in a private organization)". What is the gender of this litigant?Use names to classify the party's sex only if there is little ambiguity. | not ascertained, male - indication in opinion (e.g., use of masculine pronoun), male - assumed because of name |
| songer_appel1_7_3 | This question concerns the first listed appellant. The nature of this litigant falls into the category "natural person (excludes persons named in their official capacity or who appear because of a role in a private organization)". What is the race or ethnic identity of this litigant as identified in the opinion? | not ascertained, caucasian - specific indication in opinion, black - specific indication in opinion |
| songer_appel1_7_4 | This question concerns the first listed appellant. The nature of this litigant falls into the category "natural person (excludes persons named in their official capacity or who appear because of a role in a private organization)". What is the citizenship of this litigant as indicated in the opinion? | not ascertained, US citizen, alien |
| songer_appel1_7_5 | This question concerns the first listed appellant. The nature of this litigant falls into the category "natural person (excludes persons named in their official capacity or who appear because of a role in a private organization)". Which of these categories best describes the income of the litigant? | not ascertained, poor + wards of state, presumed poor |
| songer_appel1_8_2 | This question concerns the first listed appellant. The nature of this litigant falls into the category "miscellaneous". Which of the following categories best describes the litigant? | fiduciary, executor, or trustee, other, nature of the litigant not ascertained |
| songer_appel1_8_3 | This question concerns the first listed appellant. The nature of this litigant falls into the category "miscellaneous", specifically "fiduciary, executor, or trustee". Which of the following specific subcategories best describes the litigant? | trustee in bankruptcy - institution, trustee in bankruptcy - individual, executor or administrator of estate - institution |
| songer_appel2_1_2 | This question concerns the second listed appellant. The nature of this litigant falls into the category "private business (including criminal enterprises)". What is the scope of this business? | local, neither local nor national, national or multi-national |

| songer_appel2_1_3 | This question concerns the second listed appellant. The nature of this litigant falls into the category "private business (including criminal enterprises)". What category of business best describes the area of activity of this litigant which is involved in this case? | agriculture, mining, construction |
| --- | --- | --- |
| songer_appel2_1_4 | This question concerns the second listed appellant. The nature of this litigant falls into the category "private business (including criminal enterprises)", specifically "agriculture". What subcategory of business best describes this litigant? | single family farm, commercial farm, agri-business, farm - other |
| songer_appel2_2_2 | This question concerns the second listed appellant. The nature of this litigant falls into the category "private organization or association". What category of private associations best describes this litigant? | business, trade, professional, or union (BTPU), other |
| songer_appel2_2_3 | This question concerns the second listed appellant. The nature of this litigant falls into the category "private organization or association", specifically "business, trade, professional, or union (BTPU)". What subcategory of private association best describes this litigant? | Business or trade association, utilities co-ops, Professional association - other than law or medicine |
| songer_appel2_3_2 | This question concerns the second listed appellant. The nature of this litigant falls into the category "federal government (including DC)". Which category of federal government agencies and activities best describes this litigant? | cabinet level department, courts or legislative, agency whose first word is "federal" |
| songer_appel2_3_3 | This question concerns the second listed appellant. The nature of this litigant falls into the category "federal government (including DC)", specifically "cabinet level department". Which specific federal government agency best describes this litigant? | Department of Agriculture, Department of Commerce, Department of Defense (includes War Department and Navy Department) |
| songer_appel2_4_2 | This question concerns the second listed appellant. The nature of this litigant falls into the category "sub-state government (e.g., county, local, special district)". Which category of substate government best describes this litigant? | legislative, executive/administrative, bureaucracy providing services |

| songer_appel2_4_3 | This question concerns the second listed appellant. The nature of this litigant falls into the category "sub-state government (e.g., county, local, special district)", specifically "legislative". Which specific substate government agency best describes this litigant? | City/county council, School Board, board of trustees for college or junior college, Other legislative body |
|---|---|---|
| songer_appel2_5_2 | This question concerns the second listed appellant. The nature of this litigant falls into the category "state government (includes territories & commonwealths)". Which category of state government best describes this litigant? | legislative, executive/administrative, bureaucracy providing services |
| songer_appel2_5_3 | This question concerns the second listed appellant. The nature of this litigant falls into the category "state government (includes territories & commonwealths)", specifically "legislative". Which specific state government agency best describes this litigant? | Legislature or separate house as an organization, Legislative Committee or Commission, Other Legislative Unit |
| songer_appel2_7_2 | This question concerns the second listed appellant. The nature of this litigant falls into the category "natural person (excludes persons named in their official capacity or who appear because of a role in a private organization)". What is the gender of this litigant?Use names to classify the party's sex only if there is little ambiguity. | not ascertained, male - indication in opinion (e.g., use of masculine pronoun), male - assumed because of name |
| songer_appel2_7_3 | This question concerns the second listed appellant. The nature of this litigant falls into the category "natural person (excludes persons named in their official capacity or who appear because of a role in a private organization)". What is the race or ethnic identity of this litigant as identified in the opinion? | not ascertained, caucasian - specific indication in opinion, black - specific indication in opinion |
| songer_appel2_7_4 | This question concerns the second listed appellant. The nature of this litigant falls into the category "natural person (excludes persons named in their official capacity or who appear because of a role in a private organization)". What is the citizenship of this litigant as indicated in the opinion? | not ascertained, US citizen, alien |

| | | |
|---|---|---|
| songer_appel2_7_5 | This question concerns the second listed appellant. The nature of this litigant falls into the category "natural person (excludes persons named in their official capacity or who appear because of a role in a private organization)". Which of these categories best describes the income of the litigant? | not ascertained, poor + wards of state, presumed poor |
| songer_appel2_8_2 | This question concerns the second listed appellant. The nature of this litigant falls into the category "miscellaneous". Which of the following categories best describes the litigant? | fiduciary, executor, or trustee, other, nature of the litigant not ascertained |
| songer_appel2_8_3 | This question concerns the second listed appellant. The nature of this litigant falls into the category "miscellaneous", specifically "fiduciary, executor, or trustee". Which of the following specific subcategories best describes the litigant? | trustee in bankruptcy - institution, trustee in bankruptcy - individual, executor or administrator of estate - institution |
| songer_appfed | What is the total number of appellants in the case that fall into the category "the federal government, its agencies, and officialss"? Answer with a number. | N/A |
| songer_appfiduc | What is the total number of appellants in the case that fall into the category "fiduciaries"? Answer with a number. | N/A |
| songer_applfrom | What is the type of district court decision or judgment appealed from (i.e., the nature of the decision below in the district court)? | Trial (either jury or bench trial), Injunction or denial of injunction or stay of injunction, Summary judgment or denial of summary judgment |
| songer_appnatpr | What is the total number of appellants in the case that fall into the category "natural persons"? Answer with a number. | N/A |
| songer_appnonp | What is the total number of appellants in the case that fall into the category "groups and associations"? Answer with a number. | N/A |
| songer_appstate | What is the total number of appellants in the case that fall into the category "state governments, their agencies, and officials"? Answer with a number. | N/A |
| songer_appsubst | What is the total number of appellants in the case that fall into the category "sub-state governments, their agencies, and officials"? Answer with a number. | N/A |
| songer_attyfee | Did the court's ruling on attorneys' fees favor the appellant? | No, Yes, Mixed answer |
| songer_bank_app1 | Is the first listed appellant bankrupt? | Yes, No |
| songer_bank_app2 | Is the second listed appellant bankrupt? | Yes, No |
| songer_bank_r1 | Is the first listed respondent bankrupt? | Yes, No |

| songer_bank_r2 | Is the second listed respondent bankrupt? | Yes, No |
|---|---|---|
| songer_capric | Did the courts's use or interpretation of the arbitrary and capricious standard support the government? Note that APA allows courts to overturn agency actions deemed to be arbitrary or capricious, an abuse of discretion, or otherwise not in accordance with law. Overton Park emphasized this is a narrow standard, and one must prove that agency's action is without a rational basis. This also includes the "substantial justification" doctrine. | No, Yes, Mixed answer |
| songer_casetyp1_1-2 | What is the specific issue in the case within the general category of "issue"? | federal offense, state offense, not determined whether state or federal offense |
| songer_casetyp1_1-3-1 | What is the specific issue in the case within the general category of "issue"? | murder, rape, arson |
| songer_casetyp1_1-3-2 | What is the specific issue in the case within the general category of "issue"? | murder, rape, arson |
| songer_casetyp1_1-3-3 | What is the specific issue in the case within the general category of "issue"? | murder, rape, arson |
| songer_casetyp1_2-2 | What is the specific issue in the case within the general category of "issue"? | civil rights claims by prisoners and those accused of crimes, voting rights, race discrimination, sex discrimination, other civil rights |
| songer_casetyp1_2-3-1 | What is the specific issue in the case within the general category of "issue"? | suit for damages for false arrest or false confinement, cruel and unusual punishment, due process rights in prison |
| songer_casetyp1_2-3-2 | What is the specific issue in the case within the general category of "issue"? | voting rights - reapportionment & districting, participation rights - rights of candidates or groups to fully participate in the political process; access to ballot, voting rights - other (includes race discrimination in voting) |
| songer_casetyp1_2-3-3 | What is the specific issue in the case within the general category of "issue"? | alien petitions - (includes disputes over attempts at deportation), indian rights and law, juveniles |
| songer_casetyp1_3-2 | What is the specific issue in the case within the general category of "issue"? | religion, press, commercial, speech and other expression |
| songer_casetyp1_3-3-1 | What is the specific issue in the case within the general category of "issue"? | commercial speech, libel, slander, defamation, free exercise of religion |
| songer_casetyp1_3-3-2 | What is the specific issue in the case within the general category of "issue"? | obscenity, association, federal internal security and communist control acts, loyalty oaths, security risks |
| songer_casetyp1_4-3 | What is the specific issue in the case within the general category of "issue"? | denial of fair hearing or notice - government employees (includes claims of terminated government workers), denial of hearing or notice in non-employment context, taking clause (i.e., denial of due process under the "taking" clause of the 5th or 14th Amendments) |

| songer_casetyp1_5-3 | What is the specific issue in the case within the general category of "issue"? | abortion rights, homosexual rights where privacy claim raised, contraception and other privacy claims related to marital relations or sexual behavior (not in 501 or 502) |
|---|---|---|
| songer_casetyp1_6-3 | What is the specific issue in the case within the general category of "issue"? | union organizing, unfair labor practices, Fair Labor Standards Act issues |
| songer_casetyp1_7-2 | What is the specific issue in the case within the general category of "issue"? | taxes, patents, copyright, torts, commercial disputes |
| songer_casetyp1_7-3-1 | What is the specific issue in the case within the general category of "issue"? | state or local tax, federal taxation - individual income tax (includes taxes of individuals, fiduciaries, & estates), federal tax - business income tax (includes corporate and parnership) |
| songer_casetyp1_7-3-2 | What is the specific issue in the case within the general category of "issue"? | motor vehicle, airplane, product liability |
| songer_casetyp1_7-3-3 | What is the specific issue in the case within the general category of "issue"? | contract disputes-general (private parties) (includes breach of contract, disputes over meaning of contracts, suits for specific performance, disputes over whether contract fulfilled, claims that money owed on contract) (Note: this category is not used when the dispute fits one of the more specific categories below), disputes over government contracts, insurance disputes |
| songer_casetyp1_7-3-4 | What is the specific issue in the case within the general category of "issue"? | bankruptcy - private individual (e.g., chapter 7), bankruptcy - business reorganization (e.g., chapter 11), other bankruptcy |
| songer_casetyp1_7-3-5 | What is the specific issue in the case within the general category of "issue"? | social security benefits (including SS disability payments), other government benefit programs (e.g., welfare, RR retirement, veterans benefits, war risk insurance, food stamps), state or local economic regulation |
| songer_casetyp1_7-3-6 | What is the specific issue in the case within the general category of "issue"? | disputes over real property (private), eminent domain and disputes with government over real property, landlord - tenant disputes |
| songer_casetyp1_9-3 | What is the specific issue in the case within the general category of "issue"? | miscellaneous interstate conflict, other federalism issue (only code as issue if opinion explicitly discusses federalism as an important issue - or if opinion explicity discusses conflict of state power vs federal power), attorneys (disbarment; etc) |
| songer_casetyp2_geniss | What is the second general issue in the case, other than mainissue? | criminal, civil rights, First Amendment |
| songer_circuit | What is the circuit of the court that decided the case? | First Circuit, Second Circuit, Third Circuit |
| songer_civproc1 | What is the most frequently cited federal rule of civil procedure in the headnotes to this case? Answer with a number. | N/A |
| songer_civproc2 | What is the second most frequently cited federal rule of civil procedure in the headnotes to this case? Answer with a number. | N/A |

| songer_classact | Is the case described in the opinion as a class action suit? | No, Yes |
|---|---|---|
| songer_comment | Did the agency give proper opportunity to comment? | No, Yes, Mixed answer |
| songer_concur | What is the number of judges who concurred in the result but not in the opinion of the court? | 0, 1, 2 |
| songer_confess | Did the court conclude that a confession or an incriminating statement was improperly admitted? Consider only incriminating statements made by the defendant. | No, Yes, Yes, but error was harmless |
| songer_const1 | What is the most frequently cited provision of the U.S. Constitution in the headnotes to this case? If it is one of the original articles of the constitution, code the number of the article preceeded by two zeros. If it is an amendment to the constitution, code the number of the amendment (zero filled to two places) preceeded by a "1". Examples: 001 = Article 1 of the original constitution, 101 = 1st Amendment, 114 = 14th Amendment. | N/A |
| songer_const2 | What is the second most frequently cited provision of the U.S. Constitution in the headnotes to this case? If it is one of the original articles of the constitution, code the number of the article preceeded by two zeros. If it is an amendment to the constitution, code the number of the amendment (zero filled to two places) preceeded by a "1". Examples: 001 = Article 1 of the original constitution, 101 = 1st Amendment, 114 = 14th Amendment. | N/A |
| songer_constit | Did the court's conclusion about the constitutionality of a law or administrative action favor the appellant? | Issue not discussed, The issue was discussed in the opinion and the resolution of the issue by the court favored the respondent, The issue was discussed in the opinion and the resolution of the issue by the court favored the appellant |
| songer_counsel | Did the court rule that the defendant had inadequate counsel? | No, Yes, Yes, but error was harmless |
| songer_counsel1 | What is the nature of the counsel for the appellant? | none (pro se), court appointed, legal aid or public defender |
| songer_counsel2 | What is the nature of the counsel for the respondent? | none (pro se), court appointed, legal aid or public defender |
| songer_crmproc1 | What is the most frequently cited federal rule of criminal procedure in the headnotes to this case? Answer with a number. | N/A |
| songer_crmproc2 | What is the second most frequently cited federal rule of criminal procedure in the headnotes to this case? Answer with a number. | N/A |

| songer_crossapp | Were there cross appeals from the decision below to the court of appeals that were consolidated in the present case? | No, Yes, Not ascertained |
|---|---|---|
| songer_deathpen | Did the court conclude that the death penalty was improperly imposed? Consider only the validity of the sentence, rather than whether or not the conviction was proper. | No, Yes, Yes, but error was harmless |
| songer_decuncon | Did the court declare any statute or administrative action unconstitutional? | no declarations of unconstitutionality, act of Congress declared unconstitutional (facial invalidity), interpretation/application of federal law invalid |
| songer_denovo | Did the court's use of the standard of review, "de novo on facts" support the government? The courts generally recognize that de novo review is impractical for the bulk of agency decisions so the substantial evidence standard helps provide a middle course. Consider the de novo review of administrative action, not de novo review of trial court by appeals court. | No, Yes, Mixed answer |
| songer_direct1 | What is the ideological directionality of the court of appeals decision? | conservative, liberal, mixed |
| songer_direct2 | What is the ideological directionality of the court of appeals decision? | conservative, liberal, mixed |
| songer_discover | Did the court's interpretation of rules relating to discovery or other issues related to obtaining evidence favor the appellant? | No, Yes, Mixed answer |
| songer_dissent | What is the number of judges who dissented from the majority? | 0, 1, 2 |
| songer_district | From which district in the state was this case appealed? | Not applicable, Eastern, Western |
| songer_diverse | Did the court conclude that the parties were truly diverse? | No, Yes, Mixed answer |
| songer_dueproc | Did the interpretation of the requirements of due process by the court favor the appellant? | No, Yes, Mixed answer |
| songer_entrap | Did the court rule that the defendant was the victim of illegal entrapment? | No, Yes, Yes, but error was harmless |
| songer_erron | Did the court's use of the clearly erroneous standard support the government? That is, a somewhat narrower standard than substantial evidence, or ignoring usual agency standards. | No, Yes, Mixed answer |
| songer_execord | Did the interpretation of executive order or administrative regulation by the court favor the appellant? This does include whether or not an executive order was lawful. | No, Yes, Mixed answer |

| songer_exhaust | Did the court determine that it would not hear the appeal for one of the following reasons: a) administrative remedies had not been exhausted; or b) the issue was not ripe for judicial action? | No, Yes, Mixed answer |
|---|---|---|
| songer_fedlaw | Did the interpretation of federal statute by the court favor the appellant? | No, Yes, Mixed answer |
| songer_fedvst | Did the court rule that federal law should take precedence over state or local laws in a case involving the conflict of laws (i.e, which laws or rules apply)? | No, Yes, Mixed answer |
| songer_foreign | Did the court rule that domestic law (federal, state or local) should take precedence over foreign law in a case involving the conflict of laws (i.e., which laws or rules apply- foreign country vs federal, state, or local)? | No, Yes, Mixed answer |
| songer_freeinfo | Did the court rule in favor of the government when the administrative action in question related to the agency's providing information to those who request it? For example, Freedom of Information, issues of governmental confidentiality, or "government in the sunshine". | No, Yes, Mixed answer |
| songer_frivapp | Did the court conclude that it could not reach the merits of the case because the motion or appeal was frivolous or raised only trivial issues and was therefore not suitable for appellate review? | No, Yes, Mixed answer |
| songer_frivol | Did the court conclude that either the original case was frivolous or raised only trivial issues and therefore was not suitable for actions on the merits? | No, Yes, Mixed answer |
| songer_genapel1 | What is the nature of the first listed appellant? | private business (including criminal enterprises), private organization or association, federal government (including DC) |
| songer_genapel2 | What is the nature of the second listed appellant whose detailed code is not identical to the code for the first listed appellant? | private business (including criminal enterprises), private organization or association, federal government (including DC) |
| songer_geniss | What is the general issue in the case? | criminal, civil rights, First Amendment |
| songer_genresp1 | What is the nature of the first listed respondent? | private business (including criminal enterprises), private organization or association, federal government (including DC) |
| songer_genresp2 | What is the nature of the second listed respondent whose detailed code is not identical to the code for the first listed respondent? | private business (including criminal enterprises), private organization or association, federal government (including DC) |

| songer_genstand | Did the agency articulate the appropriate general standard? This question includes whether the agency interpreted the statute "correctly". The courts often refer here to the rational basis test, plain meaning, reasonable construction of the statute, congressional intent, etc. This issue also includes question of which law applies or whether amended law vs law before amendment applies. | No, Yes, Mixed answer |
|---|---|---|
| songer_habeas | Was the case an appeal of a decision by the district court on a petition for habeas corpus? | no, yes, state habeas corpus (criminal), yes, federal habeas corpus (criminal) |
| songer_immunity | Did the court refuse to reach the merits of the appeal because it concluded that the defendant had immunity? | No, Yes, Mixed answer |
| songer_improper | Did the court conclude that there was improper influence on the jury? For example, include jury tampering or failure to shield jury from prejudicial media accounts. Exclude prejudicial conduct by the prosecutor. | No, Yes, Yes, but error was harmless |
| songer_indict | Did the court rule that the indictment was defective? | No, Yes, Yes, but error was harmless |
| songer_indigent | Did the court rule that the defendant's rights as an indigent were violated? | No, Yes, Yes, but error was harmless |
| songer_initiate | What party initiated the appeal? | Original plaintiff, Original defendant, Federal agency representing plaintiff |
| songer_injunct | Did the court's ruling on the validity of an injunction or the denial of an injunction or a stay of injunction favor the appellant? | No, Yes, Mixed answer |
| songer_insane | Did the court below err in not permitting an insanity defense? | No, Yes, Yes, but error was harmless |
| songer_int_law | Did the court rule in favor of the appellant on an issue related to the interpretation of a treaty or international law? | No, Yes, Mixed answer |
| songer_interven | Did one or more individuals or groups seek to formally intervene in the appeals court consideration of the case? | no intervenor in case, intervenor = appellant, intervenor = respondent |
| songer_judgdisc | Did the court's ruling on the abuse of discretion by the trial judge favor the appellant? This includes the issue of whether the judge actually had the authority for the action taken, but does not include questions of discretion of administrative law judges. | No, Yes, Mixed answer |

| songer_judrev | Did the court conclude the decision was subject to judicial review? While questions of fact are subject to limited review, questions of law are subject to full review. The problem becomes determining which are clear questions of law or fact as they are often "mixed". | No, Yes, Mixed answer |
|---|---|---|
| songer_jurisdiction | Did the court determine that it had jurisdiction to hear this case? | No, Yes, Mixed answer |
| songer_juryinst | Did the court conclude that the jury instructions were improper? | No, Yes, Yes, but error was harmless |
| songer_late | Did the court refuse to decide the appeal because the appellant failed to comply with some rule relating to timeliness of the appeal? | No, Yes, Mixed answer |
| songer_majvotes | What is the number of judges who voted in favor of the disposition favored by the majority? | 0, 1, 2 |
| songer_method | What is the nature of the proceeding in the court of appeals for this case? | decided by panel for first time (no indication of re-hearing or remand), decided by panel after re-hearing (second time this case has been heard by this same panel), decided by panel after remand from Supreme Court |
| songer_mootness | Did the court conclude that an issue was moot? | No, Yes, Mixed answer |
| songer_notice | Decisions that affect life, liberty, or property must be preceded by adequate notice and an opportunity for a fair hearing. Did the agency give proper notice? | No, Yes, Mixed answer |
| songer_numappel | What is the total number of appellants in the case? Answer with a number. | N/A |
| songer_numresp | What is the total number of respondents in the case? Answer with a number. | N/A |
| songer_opinstat | Is the opinion writer identified in the opinion, or was the opinion per curiam? | Signed, with reasons, Per curiam, with reasons, Not ascertained |
| songer_origin | What type of court made the original decision? | Federal district court (single judge), 3 judge district court, State court |
| songer_othadmis | Did the court rule that some evidence, other than a confession made by the defendant or illegal search and seizure, was inadmissibile (or did ruling on appropriateness of evidentary hearing benefit the defendant)? | No, Yes, Yes, but error was harmless |
| songer_othappth | Did the court refuse to rule on the merits of the appeal because of some threshhold issue other than timeliness or frivolousness that was relevant on appeal but not at the original trial? | No, Yes, Mixed answer |

| songer_othcrim | Did the court rule for the defendant on grounds other than procedural grounds? For example, right to speedy trial, double jeopardy, confrontation, retroactivity, self defense. This includes the question of whether the defendant waived the right to raise some claim. | No, Yes, Yes, but error was harmless |
|---|---|---|
| songer_othjury | Did the court conclude that the jury composition or selection was invalid or that the jury was biased or tampered with? | No, Yes, Yes, but error was harmless |
| songer_oththres | Did the court refuse to rule on the merits of the appeal because of a threshhold issue other than lack of jurisdiction, standing, mootness, failure to state a claim, exhaustion, timeliness, immunity, frivolousness, or nonjusticiable political question? | No, Yes, Mixed answer |
| songer_plea | Did the court rule for the defendant on an issue related to plea bargaining? Plea bargain includes all challenges to plea. | No, Yes, Yes, but error was harmless |
| songer_polquest | Did the court refuse to rule on the merits of the case because it was considered to be a nonjusticiable "political question"? | No, Yes, Mixed answer |
| songer_post_trl | Did the court's ruling on some post-trial procedure or motion (e.g., allocating court costs or post award relief) favor the appellant? This doe not include attorneys' fees, but does include motions to set aside a jury verdict. | No, Yes, Mixed answer |
| songer_prejud | Was there prejudicial conduct by prosecution? | No, Yes, Yes, but error was harmless |
| songer_pretrial | Did the court's rulings on pretrial procedure favor the appellant? This includes whether or not there is a right to jury trial, whether the case should be certified as a class action, or whether a prospective party has a right to intervene in the case, but does not include rulings on motions for summary judgment. | No, Yes, Mixed answer |
| songer_procdis | Did the court uphold the dismissal by district court on procedural grounds? | No, Yes, Yes, but error was harmless |
| songer_procedur | Did the interpretation of federal rule of procedures, judicial doctrine, or case law by the court favor the appellant? | No, Yes, Mixed answer |
| songer_r_bus | What is the total number of respondents in the case that fall into the category "private business and its executives"? Answer with a number. | N/A |

| | | |
|---|---|---|
| songer_r_fed | What is the total number of respondents in the case that fall into the category "the federal government, its agencies, and officialss"? Answer with a number. | N/A |
| songer_r_fiduc | What is the total number of respondents in the case that fall into the category "fiduciaries"? Answer with a number. | N/A |
| songer_r_natpr | What is the total number of respondents in the case that fall into the category "natural persons"? Answer with a number. | N/A |
| songer_r_nonp | What is the total number of respondents in the case that fall into the category "groups and associations"? Answer with a number. | N/A |
| songer_r_state | What is the total number of respondents in the case that fall into the category "state governments, their agencies, and officials"? Answer with a number. | N/A |
| songer_r_stid | What is the state of the first listed state or local government agency that is a respondent? | not, Alabama, Alaska |
| songer_r_subst | What is the total number of respondents in the case that fall into the category "sub-state governments, their agencies, and officials"? Answer with a number. | N/A |
| songer_realapp | Are the formally listed appellants in the case the "real parties", that is, are they the parties whose real interests are most directly at stake? | both 1st and 2nd listed appellants are real parties (or only one appellant, and that appellant is a real party), the 1st appellant is not a real party, the 2nd appellant is not a real party |
| songer_realresp | Are the formally listed respondents in the case the "real parties", that is, are they the parties whose real interests are most directly at stake? | both 1st and 2nd listed respondents are real parties (or only one respondent, and that respondent is a real party), the 1st respondent is not a real party, the 2nd respondent is not a real party |
| songer_record | Did the agency fail to develop an adequate record? For example, if the court was unable to determine what doctrine was used for the decision or unable to determine the basis of the decision. | No, Yes, Mixed answer |
| songer_respond1_1_2 | This question concerns the first listed respondent. The nature of this litigant falls into the category "private business (including criminal enterprises)". What is the scope of this business? | local, neither local nor national, national or multi-national |
| songer_respond1_1_3 | This question concerns the first listed respondent. The nature of this litigant falls into the category "private business (including criminal enterprises)". What category of business best describes the area of activity of this litigant which is involved in this case? | agriculture, mining, construction |

| songer_respond1_1_4 | This question concerns the first listed respondent. The nature of this litigant falls into the category "private business (including criminal enterprises)", specifically "agriculture". What subcategory of business best describes this litigant? | single family farm, commercial farm, agri-business, farm - other |
|---|---|---|
| songer_respond1_2_2 | This question concerns the first listed respondent. The nature of this litigant falls into the category "private organization or association". What category of private associations best describes this litigant? | business, trade, professional, or union (BTPU), other |
| songer_respond1_2_3 | This question concerns the first listed respondent. The nature of this litigant falls into the category "private organization or association", specifically "business, trade, professional, or union (BTPU)". What subcategory of private association best describes this litigant? | Business or trade association, utilities co-ops, Professional association - other than law or medicine |
| songer_respond1_3_2 | This question concerns the first listed respondent. The nature of this litigant falls into the category "federal government (including DC)". Which category of federal government agencies and activities best describes this litigant? | cabinet level department, courts or legislative, agency whose first word is "federal" |
| songer_respond1_3_3 | This question concerns the first listed respondent. The nature of this litigant falls into the category "federal government (including DC)", specifically "cabinet level department". Which specific federal government agency best describes this litigant? | Department of Agriculture, Department of Commerce, Department of Defense (includes War Department and Navy Department) |
| songer_respond1_4_2 | This question concerns the first listed respondent. The nature of this litigant falls into the category "sub-state government (e.g., county, local, special district)". Which category of substate government best describes this litigant? | legislative, executive/administrative, bureaucracy providing services |
| songer_respond1_4_3 | This question concerns the first listed respondent. The nature of this litigant falls into the category "sub-state government (e.g., county, local, special district)", specifically "legislative". Which specific substate government agency best describes this litigant? | City/county council, School Board, board of trustees for college or junior college, Other legislative body |

| | | |
|---|---|---|
| songer_respond1_5_2 | This question concerns the first listed respondent. The nature of this litigant falls into the category "state government (includes territories & commonwealths)". Which category of state government best describes this litigant? | legislative, executive/administrative, bureaucracy providing services |
| songer_respond1_5_3 | This question concerns the first listed respondent. The nature of this litigant falls into the category "state government (includes territories & commonwealths)", specifically "legislative". Which specific state government agency best describes this litigant? | Legislature or separate house as an organization, Legislative Committee or Commission, Other Legislative Unit |
| songer_respond1_7_2 | This question concerns the first listed respondent. The nature of this litigant falls into the category "natural person (excludes persons named in their official capacity or who appear because of a role in a private organization)". What is the gender of this litigant? Use names to classify the party's sex only if there is little ambiguity. | not ascertained, male - indication in opinion (e.g., use of masculine pronoun), male - assumed because of name |
| songer_respond1_7_3 | This question concerns the first listed respondent. The nature of this litigant falls into the category "natural person (excludes persons named in their official capacity or who appear because of a role in a private organization)". What is the race or ethnic identity of this litigant as identified in the opinion? | not ascertained, caucasian - specific indication in opinion, black - specific indication in opinion |
| songer_respond1_7_4 | This question concerns the first listed respondent. The nature of this litigant falls into the category "natural person (excludes persons named in their official capacity or who appear because of a role in a private organization)". What is the citizenship of this litigant as indicated in the opinion? | not ascertained, US citizen, alien |
| songer_respond1_7_5 | This question concerns the first listed respondent. The nature of this litigant falls into the category "natural person (excludes persons named in their official capacity or who appear because of a role in a private organization)". Which of these categories best describes the income of the litigant? | not ascertained, poor + wards of state, presumed poor |
| songer_respond1_8_2 | This question concerns the first listed respondent. The nature of this litigant falls into the category "miscellaneous". Which of the following categories best describes the litigant? | fiduciary, executor, or trustee, other, nature of the litigant not ascertained |

| songer_respond1_8_3 | This question concerns the first listed respondent. The nature of this litigant falls into the category "miscellaneous", specifically "fiduciary, executor, or trustee". Which of the following specific subcategories best describes the litigant? | trustee in bankruptcy - institution, trustee in bankruptcy - individual, executor or administrator of estate - institution |
|---|---|---|
| songer_respond2_1_2 | This question concerns the second listed respondent. The nature of this litigant falls into the category "private business (including criminal enterprises)". What is the scope of this business? | local, neither local nor national, national or multi-national |
| songer_respond2_1_3 | This question concerns the second listed respondent. The nature of this litigant falls into the category "private business (including criminal enterprises)". What category of business best describes the area of activity of this litigant which is involved in this case? | agriculture, mining, construction |
| songer_respond2_1_4 | This question concerns the second listed respondent. The nature of this litigant falls into the category "private business (including criminal enterprises)", specifically "agriculture". What subcategory of business best describes this litigant? | single family farm, commercial farm, agri-business, farm - other |
| songer_respond2_2_2 | This question concerns the second listed respondent. The nature of this litigant falls into the category "private organization or association". What category of private associations best describes this litigant? | business, trade, professional, or union (BTPU), other |
| songer_respond2_2_3 | This question concerns the second listed respondent. The nature of this litigant falls into the category "private organization or association", specifically "business, trade, professional, or union (BTPU)". What subcategory of private association best describes this litigant? | Business or trade association, utilities co-ops, Professional association - other than law or medicine |
| songer_respond2_3_2 | This question concerns the second listed respondent. The nature of this litigant falls into the category "federal government (including DC)". Which category of federal government agencies and activities best describes this litigant? | cabinet level department, courts or legislative, agency whose first word is "federal" |
| songer_respond2_3_3 | This question concerns the second listed respondent. The nature of this litigant falls into the category "federal government (including DC)", specifically "cabinet level department". Which specific federal government agency best describes this litigant? | Department of Agriculture, Department of Commerce, Department of Defense (includes War Department and Navy Department) |

| | | |
|---|---|---|
| songer_respond2_4_2 | This question concerns the second listed respondent. The nature of this litigant falls into the category "sub-state government (e.g., county, local, special district)". Which category of substate government best describes this litigant? | legislative, executive/administrative, bureaucracy providing services |
| songer_respond2_4_3 | This question concerns the second listed respondent. The nature of this litigant falls into the category "sub-state government (e.g., county, local, special district)", specifically "legislative". Which specific substate government agency best describes this litigant? | City/county council, School Board, board of trustees for college or junior college, Other legislative body |
| songer_respond2_5_2 | This question concerns the second listed respondent. The nature of this litigant falls into the category "state government (includes territories & commonwealths)". Which category of state government best describes this litigant? | legislative, executive/administrative, bureaucracy providing services |
| songer_respond2_5_3 | This question concerns the second listed respondent. The nature of this litigant falls into the category "state government (includes territories & commonwealths)", specifically "legislative". Which specific state government agency best describes this litigant? | Legislature or separate house as an organization, Legislative Committee or Commission, Other Legislative Unit |
| songer_respond2_7_2 | This question concerns the second listed respondent. The nature of this litigant falls into the category "natural person (excludes persons named in their official capacity or who appear because of a role in a private organization)". What is the gender of this litigant?Use names to classify the party's sex only if there is little ambiguity. | not ascertained, male - indication in opinion (e.g., use of masculine pronoun), male - assumed because of name |
| songer_respond2_7_3 | This question concerns the second listed respondent. The nature of this litigant falls into the category "natural person (excludes persons named in their official capacity or who appear because of a role in a private organization)". What is the race or ethnic identity of this litigant as identified in the opinion? | not ascertained, caucasian - specific indication in opinion, black - specific indication in opinion |
| songer_respond2_7_4 | This question concerns the second listed respondent. The nature of this litigant falls into the category "natural person (excludes persons named in their official capacity or who appear because of a role in a private organization)". What is the citizenship of this litigant as indicated in the opinion? | not ascertained, US citizen, alien |

| | | |
|---|---|---|
| songer_respond2_7_5 | This question concerns the second listed respondent. The nature of this litigant falls into the category "natural person (excludes persons named in their official capacity or who appear because of a role in a private organization)". Which of these categories best describes the income of the litigant? | not ascertained, poor + wards of state, presumed poor |
| songer_respond2_8_2 | This question concerns the second listed respondent. The nature of this litigant falls into the category "miscellaneous". Which of the following categories best describes the litigant? | fiduciary, executor, or trustee, other, nature of the litigant not ascertained |
| songer_respond2_8_3 | This question concerns the second listed respondent. The nature of this litigant falls into the category "miscellaneous", specifically "fiduciary, executor, or trustee". Which of the following specific subcategories best describes the litigant? | trustee in bankruptcy - institution, trustee in bankruptcy - individual, executor or administrator of estate - institution |
| songer_rtcouns | Did the court rule that the defendant's right to counsel was violated (for some reason other than inadequate counsel)? | No, Yes, Yes, but error was harmless |
| songer_search | Did the court below improperly rule for the prosecution on an issue related to an alleged illegal search and seizure? | No, Yes, Yes, but error was harmless |
| songer_sentence | Did the court conclude that some penalty, excluding the death penalty, was improperly imposed? | No, Yes, Yes, but error was harmless |
| songer_source | What forum heard this case immediately before the case came to the court of appeals? | Federal district court (single judge), 3 judge district court, State court |
| songer_st_v_st | Did the court rule in favor of the appellant on the issue of a conflict of laws ( which laws or rules apply ) other than federal v state or foreign v domestic (e.g., one state vs second state)? | No, Yes, Mixed answer |
| songer_standing | Did the court determine that the parties had standing? | No, Yes, Mixed answer |
| songer_state | In what state or territory was the case first heard? | not, Alabama, Alaska |
| songer_stateclaim | Did the court dismiss the case because of the failure of the plaintiff to state a claim upon which relief could be granted? | No, Yes, Mixed answer |
| songer_stpolicy | Did the interpretation of state or local law, executive order, administrative regulation, doctrine, or rule of procedure by the court favor the appellant? | No, Yes, Mixed answer |

| songer_subevid | Did the court's interpretation of the substantial evidence rule support the government? For example, "such evidence as a reasonable mind might accept as adequate to support a conclusion" or "more than a mere scintilla". This issue is present only when the court indicates that it is using this doctrine, rather than when the court is merely discussing the evidence to determine whether the evidence supports the position of the appellant or respondent. | No, Yes, Mixed answer |
|---|---|---|
| songer_suffic | Did the court rule that there was insufficient evidence for conviction? | No, Yes, Yes, but error was harmless |
| songer_summary | Did the court's ruling on the appropriateness of summary judgment or the denial of summary judgment favor the appellant? | No, Yes, Mixed answer |
| songer_timely | Did the court conclude that it could not reach the merits of the case because the litigants had not complied with some rule relating to timeliness, a filing fee, or because a statute of limitations had expired? | No, Yes, Mixed answer |
| songer_treat | What is the disposition by the court of appeals of the decision of the court or agency below? | stay, petition, or motion granted, affirmed; or affirmed and petition denied, reversed (include reversed & vacated) |
| songer_trialpro | Did the court's ruling on procedure at trial favor the appellant? This includes jury instructions and motions for directed verdicts made during trial. | No, Yes, Mixed answer |
| songer_two_issues | Are there two issues in the case? | no, yes |
| songer_typeiss | What is the general category of issues discussed in the opinion of the court? | criminal and prisoner petitions, civil - government, diversity of citizenship |
| songer_usc1 | What is the most frequently cited title of the U.S. Code in the headnotes to this case? Answer with a number. | N/A |
| songer_usc1sect | What is the number of the section from the title of the most frequently cited title of the U.S. Code in the headnotes to this case, that is, title usc1? Answer with a number. | N/A |
| songer_usc2 | The most frequently cited title of the U.S. Code in the headnotes to this case is usc1. What is the second most frequently cited title of this U.S. Code in the headnotes to this case? Answer with a number. | N/A |

| songer_usc2sect | What is the number of the section from the title of the second most frequently cited title of the U.S. Code in the headnotes to this case, that is, title usc2? Answer with a number. | N/A |
|---|---|---|
| songer_weightev | Did the factual interpretation by the court or its conclusions (e.g., regarding the weight of evidence or the sufficiency of evidence) favor the appellant? | No, Yes, Mixed answer |
| songer_whlaws | Did the court's discussion of which state's laws should control their ruling in the case support the position taken by the appellant? | No, Yes, Mixed answer |

