# LAWMA:
# THE POWER OF SPECIALIZATION FOR LEGAL TASKS

## ABSTRACT

Annotation and classification of legal text are central components of empirical legal research. Traditionally, these tasks are often delegated to trained research assistants. Motivated by the advances in language modeling, empirical legal scholars are increasingly turning to commercial models, hoping that it will alleviate the significant cost of human annotation. In this work, we present a comprehensive analysis of large language models' current abilities to perform legal annotation tasks. To do so, we construct `CaselawQA`, a benchmark comprising 260 legal text classification tasks, nearly all new to the machine learning community. Starting from GPT-4 as a baseline, we show that it has non-trivial but highly varied accuracy, often exhibiting performance that may be insufficient for legal work. We then demonstrate that a lightly fine-tuned Llama 3 8B model vastly outperforms GPT-4 on almost all tasks, typically by double-digit percentage points. A few tens to hundreds of examples suffice to achieve high classification accuracy. Our work points to a viable alternative to the predominant practice of prompting commercial models. For concrete legal tasks with some available labeled data, researchers are better off using a specialized open-source model.

## 1 INTRODUCTION

The legal system generates a staggering volume of complex documents. United States federal courts alone process hundreds of thousands of cases a year, each having substantial case files. Much empirical legal research involves the systematic collection and analysis of such data in order to understand how laws function in practice and what impact they have on society. What limits researchers across the board is the cost of annotating and classifying legal documents. Legal classification tasks vary in complexity, but often require substantial expertise and effort. Employing trained research assistants stretches to a few thousand documents at a time, but is no match for the sheer scale of legal data.

There has long been an interest by empirical legal scholars in NLP tools for feature extraction (i.e., annotation) in lieu of human annotators (Livermore & Rockmore, 2019). Starting from sentiment analysis and topic models, to now large language models. The costs and error of existing methods is the single most important bottleneck in the empirical legal studies pipeline. Yet, the use of large language models to annotate legal text remains a critically understudied area.

Nonetheless, motivated by the rapid advances in language models, law scholars increasingly try out commercial models, such as GPT-4, on a variety of legal tasks, hoping to boost the efficiency of legal research. The underlying assumption is that large models such as GPT-4 provide the best solution to the problem that is currently available. In this work, we critically examine this assumption.

### 1.1 OUR CONTRIBUTIONS

We introduce and study a collection of 260 legal classification tasks, nearly all new to the machine learning community. The tasks we introduce are actual legal annotation tasks based on the U.S. Supreme Court (Spaeth et al., 2023) and Court of Appeals (Songer) databases. These databases offer rich annotations for court cases, which we utilize as labels to create challenging multi-class classification tasks. We aggregate these tasks into an easy-to-use benchmark, which we call `CaselawQA`. We detail in Section 2 the process used to construct this benchmark.

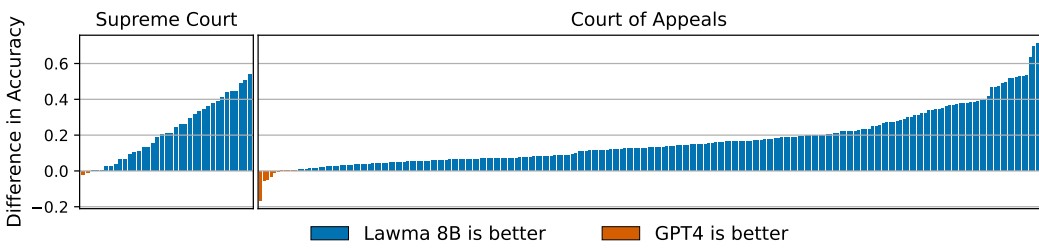

Figure 1: The cost of generality: Difference in accuracy between Lawma 8B and GPT4. Each vertical bar represents the accuracy difference on one task, sorted in ascending order.

We then evaluate in Section 3 the zero-shot performance of 28 language models, including GPT-4. We find that only a handful of them—notably Llama 3 70B Instruct and GPT-4—perform significantly better than a constant classifier that outputs the majority class. Of these models, GPT-4 delivers the strongest performance. Still, its average performance is poor (**62.0**% accuracy), and there are dozens of tasks where it performs worse than random guessing. Evaluating GPT-4 few-shot does not improve performance. Based on our comprehensive evaluations, we conclude that that the performance of current LLMs is far from sufficient for actual legal annotation work.

Next, we leverage our large corpus of legal classification tasks to fine-tune a single Llama 3 8B Instruct (MetaAI, 2024) model, which we call *Lawma 8B* (Section 4). We show that Lawma 8B achieves vastly superior performance to GPT-4[1] (Figure 1). Specifically, Lawma 8B outperforms GPT-4 by **20.0** accuracy points, attaining in absolute terms **81.9**% accuracy. Although it is expected that fine-tuning improves performance, the strong superiority of fine-tuning an open-weights model at much smaller scale is highly surprising. Our results demonstrate that, for legal classification tasks, researchers are better off using small specialized models rather than large general-purpose LLMs.

Finally, we conduct several additional large-scale fine-tuning experiments that further demonstrate the benefits and practicality of specializing models:

- Larger models respond better to fine-tuning. Across nine different base models, accuracy increases steadily with pretraining compute (Section 4.2, Figure 7). We fine-tune a single Llama 3 70B Instruct model, which we call Lawma 70B, which attains **83.3**% accuracy.

- Fine-tuning is data efficient. A few hundred examples typically suffice to achieve high classification accuracy (Section 4.3, Figure 8). This is crucial, since labeling a few hundred data points is often financially feasible for many legal scholars.

- Fine-tuning generalizes to unseen tasks. Fine-tuning only on Court of Appeals tasks improves its accuracy on Supreme Court tasks by 18.8 accuracy points (Section 4.4, Figure 9).

- We can simultaneously fine-tune on all 260 tasks. There is not a large loss compared with fine-tuning on a single specific task (Appendix D, Figure 11). This is desirable in practice, as it obviates the need to train and maintain a separate model for each task.

Our results speak to the power of specialization for legal classification tasks. The methods described in our paper can radically expand the capacity of legal scholars to engage in quantitative work, empowering legal scholars to apply the "law as data" paradigm to a host of novel research questions. Annotations of existing datasets can become much more fine-grained. Entire jurisdictions that have hitherto escaped academic attention, such as the many courts of U.S. States, may finally be analyzed.

From a benchmarking perspective, the tasks presented in this work are of independent interest. They are challenging multi-class classification problems that require some amount of legal expertise. The best models achieve non-trivial, but modest zero-shot performance. And even fine-tuned models don't reach intercoder agreement rates (Section C). Our empirical findings suggest that these legal classification tasks are diverse, non-trivial evaluation tasks for future model advances.

---

[1]We evaluate `gpt-4-0613`, which is what at the time of writing `gpt-4` points to in the API. The recently released GPT-4o and o1 models are currently not available for our region via the Azure OpenAI Service.

Finally, our work challenges the prevailing narrative about the suitability of "generalist" models. In commercial APIs, users are generally limited to prompting generalist models, as fine-tuning is costly for the model provider. But as we show, generalist models are neither sufficiently good nor best possible for many practical tasks. Specializing models to concrete tasks of interests, even using relatively few labeled examples, can provide a simple, practical, and far more accurate solution.

## 1.2 RELATED WORK

**Benchmarks for legal tasks.** LegalBench (Guha et al., 2023) is a recent multi-task benchmark for natural language understanding in legal domains. As of writing, LegalBench consists of 162 tasks gathered from 40 contributors. LegalBench draws on numerous earlier benchmarking efforts in different legal domains, specifically, inference on contracts (Koreeda & Manning, 2021; Hendrycks et al., 2021), merger agreement understanding (Wang et al., 2023), identifying the legal holding of a case (Zheng et al., 2021), statutory reasoning (Holzenberger & Van Durme, 2021), privacy compliance and policy (Wilson et al., 2016; Zimmeck et al., 2019; Ravichander et al., 2019), and identifying unfair clauses in terms of service (Lippi et al., 2019). Bhambhoria et al. (2024) evaluate the performance of general-purpose models on legal question-answering tasks and advocate for the development of open-source models tailored to the legal domain. We extend and strengthen these valuable efforts to benchmark large language models in legal settings. We focus on core legal classification tasks based on the U.S. Supreme Court Database (Spaeth et al., 2023) and the U.S. Courts of Appeals database (Songer). Our evaluation suite measures the performance of models in annotating court opinions, focusing on tasks that are of interest to the field of empirical legal studies. The tasks we study are complementary to those in LegalBench. We do not evaluate our model on LegalBench, since our model is specialized to the Supreme Court and Appeals Court data.

**Large language models for the legal domain.** General-purpose language models are likely to be trained on a substantial amount of legal data because much of this data is publicly available on the internet. For example, the FreeLaw dataset includes a large collection of court opinions (Gao et al., 2021). Legal-BERT (Chalkidis et al., 2020) is a BERT-like transformer model that was pretrained on a few hundred thousand legal documents. The more recent SaulLM models (Colombo et al., 2024b;a) adapt the open-weights Mistral (Jiang et al., 2023; 2024) models to the legal domain both by continual pretraining and instruction-tuning on legal text. In contrast to Lawma, we consider SaulLM to be a general-purpose model for the legal domain, not tailored to any specific legal task. Our approach differs significantly; we focus on developing models specialized for annotation tasks of practical interest to empirical legal studies. We demonstrate that specialization is highly effective, with our Lawma models significantly outperforming all other evaluated LLMs. For a discussion on the adoption of large language models in the legal community, refer to Appendix A.

**Data annotation and labeling.** Hall & Wright (2008) provide an overview of the use of human annotators in empirical legal studies. Student coders have been deployed to extract a wide variety of features from legal data. Although student researchers are much less expensive than private attorneys, the costs can quickly become prohibitive. Depending on the size of the document and the complexity of the task, research assistants can label roughly dozens of examples per hour. Projects involving the labeling of hundreds of documents are financially feasible for many legal scholars, but projects involving many thousands of documents are largely impractical. In an example of a larger annotation effort, Frankenreiter et al. (2021) employed human coders to annotate several thousands of corporate charters. Using ChatGPT for a similar task, Frankenreiter & Talley (2024) estimated that employing human coders would have been approximately ten times more costly.

Data annotation and labeling also play a major role in machine learning benchmarks and applications, see, e.g., Aroyo & Welty (2015); Gray & Suri (2019); Hardt & Recht (2022) for background. Dorner & Hardt (2024) give an extended discussion about label quality and annotator disagreement in the context of machine learning benchmarks.

## 1.3 LIMITATIONS

Fine-tuning increases accuracy to about 80% in our evaluation suite compared with around 60% for non-specialized models. While we are rather certain that 60% accuracy is insufficient for consequential legal work, we emphasize that 80% is still far from perfect. In addition, the variance

```
What follows is an opinion from the Supreme Court of the United States.
Your task is to identify  whether the opinion effectively says that the
decision in this case  "overruled" one or more of the Court\'s own
precedents. Alteration also extends to language in the majority opinion
that states that a precedent of the Supreme Court  has been "disapproved,"
or is "no longer good law". Note, however, that alteration does not
apply to cases in which the Court "distinguishes" a precedent.

[COURT OPINION]

Question: Did the the decision of the court overrule one or more of the
Court's own precedents?
A. Yes
B. No
Answer:
```

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

 start by evaluating the performance of different language models on the extended test set of `CaselawQA`. We choose language models that are of particular relevance to the legal domain: LegalBert (Chalkidis et al., 2020), as well as SaulLM 7B (Colombo et al., 2024b), its base model Mistral 7B Instruct (Jiang et al., 2023), and its Mixture-of-Experts variant Mixtral 8x7B (Jiang et al., 2024). We additionally evaluate GPT-4 (Achiam et al., 2023) due to its prevalent use among legal scholars, and the Llama 3 Instruct (MetaAI, 2024) models, which are arguably the best performing open-weights models at present time. We report the evaluation results in Figure 4. We include as baseline the majority classifier which simply outputs the majority class of each classification tasks.

Despite the popularity of LegalBERT, we observe that it performs worse than the majority classifier baseline. This is unsurprising, as by current standards it has both a small size (110M parameters) and a small context window (256 tokens). SaulLM 7B, the other legal-domain model, similarly fails to beat the majority classifier baseline. In fact, SaulLM 7B underperforms compared to its base model Mistral 7B Instruct both across all tasks and Court of Appeals tasks. This indicates that broadly adapting models to the legal domain may not prove beneficial for annotation work. Overall, we find that existing LLMs tailored for the legal domain obtain trivial performance in our annotation tasks.

In fact, we observe that only the two largest models tested, Llama 3 70B Instruct and GPT-4, perform substantially better than the majority classifier baseline. Still, their performance is modest ($< 65\%$ accuracy), and there are dozens of tasks where both models perform worse than random guessing, see Figure 16 in Appendix E. Our evaluations therefore indicate that, for most tasks, the performance of general-purpose LLMs is clearly insufficient for consequential legal annotation work.

---

[2]Note that more involved prompting strategies –e.g., chain-of-thought (Wei et al., 2022)– can yield better task performance but are substantially more expensive. For legal tasks specifically, the choice of prompt can have a significant effect in performance (Li et al., 2024). However, prompt tuning requires task-specific domain knowledge and can be reasonably time consuming.

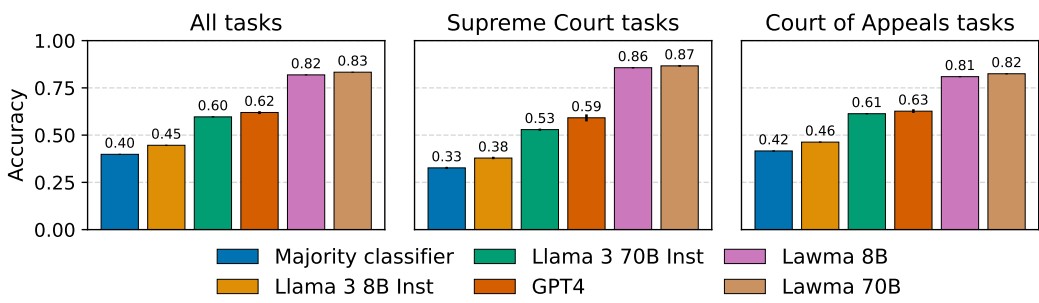

Figure 5: Accuracy of the Lawma models on the extended test set of `CaselawQA`.

**Few-shot evaluation**   We also consider whether evaluating GPT-4 few-shot leads to any improvements. Whereas the default context window for GPT-4 is 8,000 tokens, a version with 32,000 tokens is available at twice the cost. We evaluated the 32k version with 3-shot prompting, since it is often unfeasible to fit more than 3 task examples within the 32k context window. Labeling each example 3-shot is $3 \times 2 = 6$ times more expensive compared to the zero-shot GPT-4 evaluation. To compensate for the increase in cost, we evaluate the model 3-shot on roughly 5% of the test examples compared those used to evaluate GPT-4. Our evaluation shows that GPT-4's performance with 3-shot prompting does not improve over zero-shot prompting, as detailed in Table 1. This is likely because most legal classification tasks involve more than three classes, meaning that three in-context examples do not cover all possible answer choices. Consequently, the model often responds with one of the three presented examples, leading to a significant drop in performance. Few-shot prompting is therefore not a fruitful strategy to adapt the model to the legal classification tasks at hand.

Table 1: Zero-shot and few-shot accuracy of GPT-4.

| Model | All tasks | Supreme Court | Court of Appeals |
|---|---|---|---|
| GPT-4 zero-shot | $62.0 \pm 0.4$ | $59.2 \pm 0.8$ | $62.7 \pm 0.5$ |
| GPT-4 $32k$ 3-shot | $60.4 \pm 1.9$ | $50.5 \pm 4.3$ | $62.9 \pm 2.1$ |

## 4   FINE-TUNING AND THE POWER OF SPECIALIZATION

In this section, we present a detailed analysis of how models can be specialized for legal classification tasks. We start by fine-tuning Llama 3 8B Inst and Llama 3 70B Inst on all 260 tasks simultaneously, resulting in our Lawma 8B and Lawma 70B models. We then perform additional fine-tuning experiments highlighting different aspects, including the scaling behaviour of fine-tuning, its sample efficiency, and its generalization to unseen tasks and Courts.

### 4.1   THE LAWMA MODELS

We first leverage our large corpus of legal classification tasks to fine-tune Llama 3 8B Instruct and Llama 3 70B Instruct on *all tasks* simultaneously. We refer to these fine-tuned models as Lawma 8B and Lawma 70B, respectively. We fine-tune on the 260 classification tasks described in Section 2.2. The fine-tuning dataset contains a total of 503,698 task examples and 1.96B tokens. See Appendix F for additional details regarding the model fine-tuning.

We compare in Figure 5 the task accuracies of Lawma 8B and Lawma 70B to that of their respective base models Llama 3 8B Instruct and Llama 3 70B Instruct, as well as GPT-4. Fine-tuning leads to remarkably large improvements in average task accuracy: Lawma 8B outperforms Llama 3 8B Instruct by **37.2** accuracy points and Lawma 70B outperforms Llama 3 70B Instruct by **21.3** accuracy points. Both Lawma 8B and Lawma 70B outperform GPT-4, Lawma 8B by **19.9** accuracy points and Lawma 70B by **21.3** accuracy points. In fact, both Lawma 8B and Lawma 70B outperform GPT-4 in about 95% of all tasks, see Figure 1. Figure 6 further demonstrates the large effect of fine-tuning by showing the histogram of task accuracies of Lawma in comparison with Llama 3 and GPT-4.

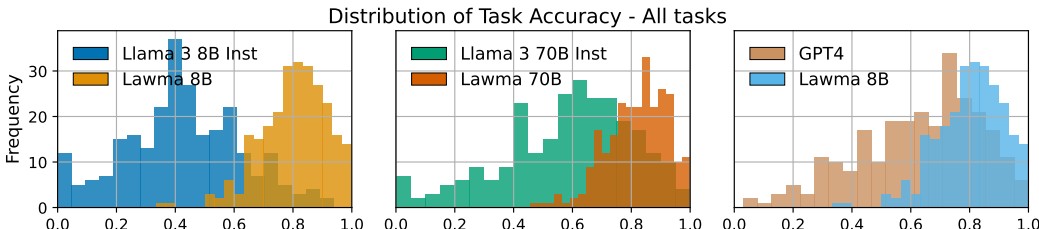

Figure 6: Distribution of task performance across all tasks for Llama 3, GPT4, and Lawma.

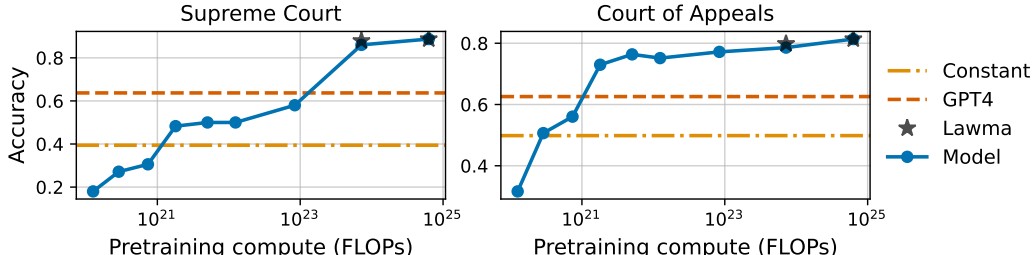

Figure 7: Performance after one epoch of fine-tuning increases monotonically in the amount of pretraining of the base model. Models left to right (blue dots): Pythia 70M, Pythia 160M, Pythia 410M, Pythia 1B, Pythia 2.8B, Pythia 6.9B, Llama 2 7B, Llama 3 8B Inst, Llama 3 70B Inst.

We find that Lawma 8B closely matches the performance of Lawma 70B. Specifically, Lawma 70B outperforms Lawma 8B by only 1.0 accuracy points for Supreme Court tasks and by 1.5 accuracy points for Appeals Court tasks (Figure 5). This suggests that, for our fine-tuning dataset, further scaling model size (e.g., fine-tuning GPT-4) is unlikely to yield major improvements. A more promising direction is to instead improve the diversity and quantity of the fine-tuning data. On the flip side, practitioners may choose to use Lawma 8B instead of the 70B model with little cost in performance.

## 4.2 Performance after fine-tuning scales with pretraining compute

The performance of specialized models tends to scale with pretraining compute (Dominguez-Olmedo et al., 2024). We investigate how performance after fine-tuning scales with the pretraining compute of the base model. We fine-tune the following models for a single epoch: Pythia 70M, Pythia 160M, Pythia 410M, Pythia 1B, Pythia 2.8B, Pythia 6.9B (Biderman et al., 2023), Llama 2 7B (Touvron et al., 2023), Llama 3 8B Instruct and Llama 3 70B Instruct. We fine-tune on all 260 tasks simultaneously. We approximate pretraining compute in FLOPs as $C \approx 6 \cdot N \cdot D$ (Kaplan et al., 2020), where $N$ is model size and $D$ is the number of pretraining tokens.

We find that mean task accuracy after fine-tuning improves with pretraining compute (Figure 7). However, we find signs of diminishing returns. For the Supreme Court tasks, performance increases steadily from $10^{20}$ to $10^{24}$ FLOPs, but further scaling to $10^{25}$ FLOPs only improves performance by an additional 4.0 accuracy points. For Appeals Court tasks, performance sharply increases from $10^{20}$ to $10^{21}$ FLOPs (i.e., Pythia 1B – which interestingly already beats GPT-4 zero-shot), but stagnates thereafter, only improving by an additional 8.5 accuracy points when scaling to $10^{25}$ FLOPs.

Our findings suggest that major improvements will likely not come from model scale alone. Rather, future work should focus on obtaining better scaling behavior. One promising direction is to improve the quality, quantity and diversity of the fine-tuning data.

## 4.3 Sample efficiency

We study how task accuracy scales as models fine-tune on more training examples. We consider the 10 tasks highlighted in Section B. We fine-tune Llama 3 8B Instruct on each task independently, rather than on all tasks simultaneously as in the previous experiments. For each task, we fine-tune

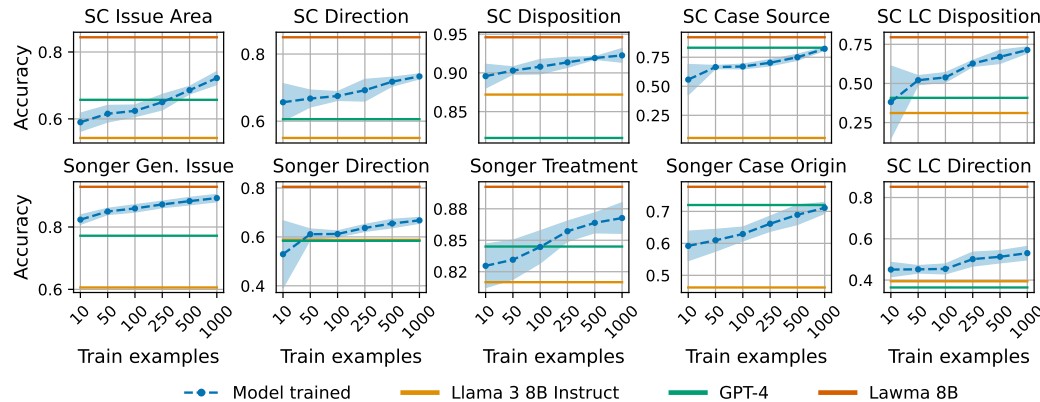

Figure 8: Sample efficiency of fine-tuning Llama 3 8B on a single task. Hundreds of task examples are typically enough to match or beat the zero-shot performance of GPT-4. Dashed blue line indicates the accuracy of Llama 3 8B fine-tuned on a single task as a function of the number of training examples. The shaded area indicates the 95% confidence interval over 5 random seeds.

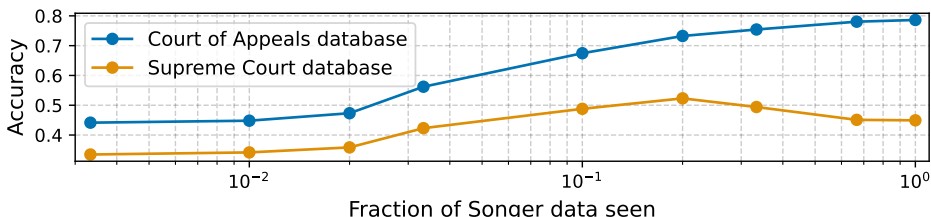

Figure 9: Fine-tuning on the Court of Appeals tasks improves accuracy on Supreme Court tasks.

on 10, 50, 100, 250, 500, and 1000 task examples. We select task examples uniformly at random, and train 5 different models corresponding to different random seeds on the examples selected for training. We therefore fine-tune and evaluate a total of $10 \cdot 6 \cdot 5 = 300$ models. We fine-tune for a maximum of 20 epochs and early stop when validation loss increases for 3 consecutive epochs.

Figure 8 shows how accuracy improves with the number of training examples. Fifty training examples are enough to match or beat the GPT-4 zero-shot baseline for 6 out of the 10 highlighted tasks, and 250 traning examples are enough to match or beat GPT-4 for 8 out of the 10 highlighted tasks. This is crucial, since labeling a few hundred data points is often financially feasible for many legal scholars (Hall & Wright, 2008). With relative few labelled task examples, fine-tuning reasonably small publicly available models can be competitive with state-of-the-art closed models. Moreover, accuracy continues to improve significantly with additional examples. With one thousand training examples, fine-tuning Llama 3 8B Inst matches or beats the GPT-4 baseline for all highlighted tasks.

## 4.4 GENERALIZATION TO UNSEEN DATABASES

We now investigate whether fine-tuning only on the Songer Appeals Court database allows us to generalize to the Supreme Court database. We fine-tune Llama 3 8B Inst for one epoch on all Songer tasks simultaneously. We plot in Figure 9 the mean accuracy for Court of Appeals tasks and Supreme Court tasks at intermediate checkpoints. As expected, performance on Court of Appeals tasks improves monotonically with the number of training examples seen. More interestingly, we observe that mean task accuracy for the Supreme Court also improves substantially, by up to 18.8 ac-

curacy points at 20% of the training steps[3]. Thereafter, performance degrades, seemingly plateauing at 11.3 accuracy points above the baseline non-finetuned performance of Llama 3 8B Inst.

Our findings indicate that, since there is some degree of overlap between Court of Appeal and Supreme Court tasks, fine-tuning on the former transfers to the latter. This suggests that Lawma might be of practical use beyond the Supreme Court and Court of Appeals tasks it was trained on.

Note, however, that fine-tuning only on the Court of Appeals database results in a mean case accuracy of 51.6%, compared to 82.4% for Lawma 8B. That is, not fine-tuning on Supreme Court cases results in a 30.9 accuracy points decrease in performance. These results again highlight the importance of fine-tuning precisely on the target tasks of interest.

## 5 DISCUSSION

We introduce and study a collection of 260 legal classification tasks, nearly all new to the machine learning community. CaselawQA, our introduced dataset, serves a double purpose: a benchmark to evaluate the ability of LLMs to perform legal annotation work of practical interest to legal scholars, and a fine-tuning dataset to specialize existing models to such legal classification tasks.

As we show, the performance of existing "generalist" LLMs is far from sufficient for consequential legal annotation work. In contrast, we demonstrate the power of specialization: we fine-tune and make available the Lawma 8B and Lawma 70B models, which strongly outperform all other models evaluated, including GPT-4 and two existing legal-domain LLMs.

The CaselawQA dataset, the Lawma models, and more broadly the specialization methodology presented in this work, are all of practical interest to legal research. The cost of human annotators represents a considerable bottleneck for the field of empirical legal studies. The advent of low-cost and flexible tools for data extraction can lead to tremendous boosts in scholarly productivity and knowledge production. For example, the falling cost of genetic sequencing led to a paradigm shift across the biological sciences, as genetic data became increasingly available in fields as disparate as public health and entomology (Köser et al., 2012; Ballare et al., 2019). A flexible automated feature extraction tool for legal texts holds similar potential for empirical legal studies, as a large realm of conceivable but impractically expensive research projects becomes accessible.

The tasks we introduce are also interesting from a broader LLM benchmarking perspective. The accuracy numbers are neither too low nor too high. The best models achieve non-trivial, but modest zero-shot performance. And even fine-tuned models don't reach intercoder agreement rates. This situation suggests that these legal classification tasks may be good test cases for future model advances. As such, we hope to extend and strengthen existing evaluation efforts.

Lastly, our work challenges the prevailing narrative about the suitability of "generalist" models. The generalist abilities of large language models are vital for commercial APIs, where users are largely restricted to prompting. But as we show, generalist models may be neither sufficiently good nor the best possible solution for many practical tasks.

We show that this is certainly the case for annotation work that arises in empirical legal research.