# OpenReview forum: "Lawma: The Power of Specialization for Legal Annotation"
_ICLR.cc/2025/Conference — ICLR 2025 Poster_

### Official Review · Reviewer_LXJY · 2024-10-30

**Soundness:** 3
**Presentation:** 3
**Contribution:** 2
**Rating:** 6
**Confidence:** 4

**Summary:**

The paper explores the use of specialized language models for legal text classification, introducing CaselawQA, a dataset with 260 legal classification tasks based on U.S. Supreme Court and Court of Appeals cases. The authors evaluate various models, including GPT-4 and a fine-tuned LLaMA model (Lawma), to test whether specialization improves performance. They find that Lawma, trained specifically on CaselawQA, outperforms GPT-4 in accuracy by up to 20 percentage points. This performance boost highlights that fine-tuned, domain-specific models can handle nuanced legal tasks more effectively than general-purpose models. Additionally, the authors demonstrate that Lawma achieves high accuracy with limited labeled data, making it a feasible and cost-effective option for empirical legal research.

**Strengths:**

- **Clarity and Quality of Writing:** The paper is well-written, with clear explanations of the methodology and findings.
- **Dataset Contribution (CaselawQA):** CaselawQA is a valuable addition to the legal NLP field, featuring a comprehensive set of 260 legal classification tasks that reveal the limitations of general-purpose models in handling specialized legal contexts. Its diverse structure makes CaselawQA a beneficial resource for benchmarking capabilities of LLMs and advancing specialized model development.
- **Empirical Validation of Fine-tuning:** The study demonstrates the advantages of fine-tuning open-source models for legal tasks, with Lawma, a fine-tuned Llama model, consistently outperforming larger general-purpose LLMs like GPT-4 across various legal tasks. This empirical evidence supports the claim that domain-specific fine-tuning can yield more effective models for niche tasks, particularly within the legal domain.
- **Sample Efficiency Insights:** By analyzing performance across different sample sizes, the study provides valuable insights into the sample efficiency of fine-tuning, showing that Lawma can achieve high accuracy with limited labeled data. This practical approach underscores the feasibility of creating specialized models even with restricted datasets, a crucial benefit for resource-limited legal research settings.

**Weaknesses:**

- **Limited Novelty in Contribution:** Although the paper effectively demonstrates the value of task-specific fine-tuning, the concept itself is not novel. Prior work has already shown that fine-tuned, specialized models outperform general-purpose LLMs [1, 2, 3]. This paper reinforces existing ideas rather than offering new methodologies or innovations in model specialization.

- **Overlooked Generalizability:** By evaluating Lawma exclusively on CaselawQA, a U.S.-specific dataset, the study limits insights into its effectiveness across other legal systems and domains [4, 5]. Without comparisons on datasets from different jurisdictions or areas of law, it’s challenging to assess the model’s broader applicability. Expanding evaluations to diverse legal datasets could strengthen the paper’s general claims about the effectiveness of fine-tuning for legal NLP tasks.

- **Narrow Focus on Text Classification:** The paper’s focus is restricted to text classification tasks, a valuable but limited subset of legal NLP. Broader experimentation on tasks such as legal text summarization for real-world low-resource data and hardware scenarios [6] could enhance the impact and relevance of the Lawma contribution for the legal community.

- **English-only Focus:** The experiments are conducted solely in English, overlooking the need for legal NLP solutions in low-resource languages and multilingual contexts [7, 8]. Legal research frequently requires cross-linguistic and multilingual analysis, and models that generalize across languages would have broader utility.

**Minor Presentation Weaknesses:**
- Typographical errors (e.g., "we conclude **that that** the performance") need correction.
- Figure 1 lacks clarity and could benefit from improved labeling or explanation to make differences in model performance more interpretable.
- The Limitations section would be more effective if positioned after the Conclusion, rather than within the Introduction, to ensure that findings are fully contextualized before limitations are addressed.


**References:**
1. UniversalNER: Targeted Distillation from Large Language Models for Open Named Entity Recognition. ICLR 2024.
2. Distilling Step-by-Step! Outperforming Larger Language Models with Less Training Data and Smaller Model Sizes. ACL 2023.
3. Fine-Tuned 'Small' LLMs (Still) Significantly Outperform Zero-Shot Generative AI Models in Text Classification. arXiv 2024.
4. LAWSUIT: a LArge expert-Written SUmmarization dataset of ITalian constitutional court verdicts. Artificial Intelligence and Law 2024.
5. Applicability of Large Language Models and Generative Models for Legal Case Judgement Summarization. Artificial Intelligence and Law 2024.
6. Semantic Self-Segmentation for Abstractive Summarization of Long Documents in Low-Resource Regimes. AAAI 2022.
7. MultiEURLEX - A Multi-Lingual and Multi-Label Legal Document Classification Dataset for Zero-Shot Cross-Lingual Transfer. EMNLP 2021.
8. Multi-Language Transfer Learning for Low-Resource Legal Case Summarization. Artificial Intelligence and Law 2023.

**Questions:**

- **Generalization to Other Jurisdictions:** Can the authors provide insights or preliminary findings on how well the Lawma models might transfer to legal data from jurisdictions outside the U.S. or to other branches and tasks of law?
- **Impact of Fine-tuning on Error Patterns:** What are some examples of typical errors Lawma makes, especially on complex tasks? Understanding these errors could help refine the approach and better inform future model improvements.
- **Exploration of Advanced Prompting Techniques:** The paper mentions that few-shot prompting did not improve GPT-4’s performance significantly. Were other advanced prompting methods, such as chain-of-thought or other reasoning prompts, considered? This could be relevant, as legal tasks often benefit from reasoning-style responses.

---

> ### Author Response · Authors · 2024-11-26
>
> Thank you for your thoughtful review.
>
> > Limited Novelty in Contribution: Although the paper effectively demonstrates the value of task-specific fine-tuning, the concept itself is not novel. Prior work has already shown that fine-tuned, specialized models outperform general-purpose LLMs [1, 2, 3].
>
> Frontier LLMs are state-of-the-art in many domains, such as mathematical reasoning (e.g., MATH), code (e.g., HumanEval), reasoning over text (e.g., DROP), and graduate-level question answering (e.g., GPQA). Prior to our work, it was not at all evident that specialized open-weights models could match the performance of frontier models on legal annotation, let alone substantially outperform them. It is precisely for this reason that recent works that use LLMs for legal annotation rely heavily on frontier models (e.g, [1, 2, 3, 4]).
>
> Please note that the references provided focus on more “classical” NLP tasks such as named entity recognition, natural language inference, or sentiment analysis. We will make sure to cite and discuss the provided references in the related work.
>
> [1] Saromir Savelka and Kevin D Ashley. The unreasonable effectiveness of large language models in zero-shot semantic annotation of legal texts. Frontiers in Artificial Intelligence, 6, 2023.
>
> [2] Michael A Livermore, Felix Herron, and Daniel Rockmore. Language model interpretability and empirical legal studies. Virginia Public Law and Legal Theory Research Paper, (2023-69), 2023.
>
> [3] Jens Frankenreiter and Eric L Talley. Sticky charters? the surprisingly tepid embrace of officer-protecting waivers in delaware. European Corporate Governance Institute-Law Working Paper, (762), 2024
>
> [4] Morgan A Gray, Jaromir Savelka, Wesley M Oliver, and Kevin D Ashley. Empirical legal analysis simplified: reducing complexity through automatic identification and evaluation of legally relevant factors. Philosophical Transactions of the Royal Society A, 382(2270):20230155, 2024.
>
> > Exploration of Advanced Prompting Techniques
>
> We have now evaluated the models using zero-shot chain of thought (CoT). We follow the standard methodology of eliciting CoT by appending to the prompt “Let’s think step by step.” Since CoT requires two orders of magnitude more compute for evaluation than the standard QA approach, we only evaluate Llama 3 8B Instruct and Llama 3 70B Instruct. This required over 500 H100 GPU hours. We observe that CoT leads to modest improvements of performance for both the 8B and 70B model, on average of 2 to 3 accuracy points, see Figure 17 in Appendix E.5. Nonetheless, Lawma 8B still strongly outperforms Llama 3 70B, by over 20 accuracy points.
>
> > Narrow Focus on Text Classification
>
> We study precisely the narrow specialization of models for legal annotation. This is a feature of work. We show that specialized models strongly outperform generalist models such as GPT-4. Such narrow specialized models have rich scientific applications for empirical legal studies and the broader “law as data” research paradigm. Entire subfields of legal scholarship, political science, economics and sociology build on law as data. Other NLP tasks (i.e. text generation) are very much of secondary importance for this domain of research.
>
> > What are some examples of typical errors Lawma makes, especially on complex tasks?
>
> The types of errors made by Lawma are highly task dependent, and thus it is difficult to draw broad conclusions. Let us give one illustrative example of a failure case. For the Supreme Court issue area classification, Lawma tends to miss classify habeas corpus cases as “Criminal Procedure” rather than “Civil rights” cases, since the language of habeas corpus cases tends to be more similar to that of criminal cases. More broadly, we find that Lawma tends to excel in tasks for which only surface correlations (e.g., the “language” of the case) suffice for accurate prediction, but struggles for tasks that require deeper understanding of the substantive aspects of the case. Lawma also tends to perform worse for tasks that have fewer training data and larger number of classes.

---

> ### Author Response · Authors · 2024-11-26
>
> > Overlooked Generalizability: [...] Without comparisons on datasets from different jurisdictions or areas of law, it’s challenging to assess the model’s broader applicability. [...] Can the authors provide insights or preliminary findings on how well the Lawma models might transfer to legal data from jurisdictions outside the U.S. or to other branches and tasks of law?
>
> We study how much task-specific fine-tuning might generalize in Section 4.4 and in particular Figure 9. We observe that fine-tuning only on the Court of Appeals database results in a mean case accuracy of 51.6%, compared to 82.4% for Lawma 8B (Section 4.4). That is, not fine-tuning on Supreme Court leaves a lot of accuracy on the table, 30.9 accuracy points to be precise. It is plausible that considering different jurisdictions or other branches of the law might result in even more stark results.
>
> Our recommendation is therefore to always specialize on the particular legal tasks of interest, as performance may otherwise be poor. Fortunately, hundreds of labelled examples are often sufficient to obtain large performance gains (Section 4.3). Thus, for large-scale data annotation the following strategy may be highly beneficial: collect a few hundred labeled examples using human annotators, fine-tune an open-weights model, and use the fine-tuned model to annotate the remaining cases at scale.

---

> > ### Comment · Reviewer_WAb5 · 2024-11-26
> > **Response**
> >
> > I have updated my score.

---

> > ### Comment · Reviewer_LXJY · 2024-11-28
> > **Aknowledgement**
> >
> > Thanks for the replies. Hoping that you will improve the manuscript accordingly with all the suggestions and discussions listed, i have increased my score.

---

### Official Review · Reviewer_VqTU · 2024-11-01

**Soundness:** 4
**Presentation:** 4
**Contribution:** 3
**Rating:** 8
**Confidence:** 4

**Summary:**

This paper proposes a new dataset of 260 classification tasks, derived from the USDB and USCAD resources. These tasks each require fairly long contexts, and can have a large number of classes (e.g. multi-label classification). They combine these tasks into an evaluation benchmark and show results for a variety of legal and otherwise SoTA models.  They find that most models struggle, although performance follows a monotonically increasing pattern based on pre-training compute.

They then propose to fine-tune the model on these law tasks, and show that fine-tuning them (which they call Lawma) results is significantly improved performance, across a wide range of tasks.

**Strengths:**

- The data proposed seems very useful and covers a broad range
- The analysis evaluates a wide range of models and situations
- The fine-tuning experiments show large gains can be had with domain-specific specialization.
- The appendix has a lot of good information about where the data came from and their inter annotator agreements

**Weaknesses:**

- It is somewhat unclear how the authors created these tasks, in terms of how the questions were designed. E.g. who wrote the explanation for each of the legal variables provided by USCAD? How are the authors sure that these are accurate representations of the classification?
- [Minor] some of these tasks are pretty niceh/easy ("What state is associated with the respondent") but again this comes from using the variables in some schema.

**Questions:**

From Weakness 1: How did you make the prompts for the questions?
Q2: What license does this fall under and will it be publicly released?

---

> ### Author Response · Authors · 2024-11-26
>
> Thank you for your thoughtful and positive review.
>
> > How did you make the prompts for the questions? Who wrote the explanation for each of the legal variables provided by USCAD? How are the authors sure that these are accurate representations of the classification?
>
> The prompts follow the MMLU multiple-choice question answering style that is popular for LLM evaluations. For the general description of each task, we take the codebook’s (either SCDB or USCAD) description of the corresponding variable in the database. We make only very minor modifications to fit an instruction style (e.g., “This field identifies the forum that heard this case immediately before the case came to the court of appeals.” -> “Your task is to identify the forum that heard this case immediately before the case came to the court of appeals.”). Since the descriptions are directly taken from the databases’ extensive codebooks, we are sure that they accurately reflect the underlying legal annotation tasks.
>
> > What license does this fall under and will it be publicly released?
>
> The benchmark, fine-tuning dataset, model weights, and all code used to construct CaselawQA and fine-tune the Lawma models are publicly available under an MIT License; however, they cannot be directly linked here to preserve anonymity.

---

### Official Review · Reviewer_WAb5 · 2024-11-03

**Soundness:** 3
**Presentation:** 2
**Contribution:** 2
**Rating:** 6
**Confidence:** 4

**Summary:**

The authors introduce a dataset of 260 legal text classification tasks, by extracting case text and labels from US Supreme Court and Court of Appeals. They evaluate different approaches such as prompt engineering, few-shot learning, and fine-tuning. It is not clear how much engagement the authors had with true legal experts in the creation of this dataset.

**Strengths:**

- Legal text processing has economic value but is difficult for most state of the art LLMs.
- The authors assemble a large dataset of real-world legal documents (via querying 3rd party services), annotated with diverse question-answering tasks.
- They show the effectiveness of fine-tuning on this dataset, over few-shot learning. They report performance of various tuning configurations.

**Weaknesses:**

- There is little exploration of the relative difficulty of the different types of tasks they introduce—it would be useful to know more about which tasks that GPT-4 outperformed on, and where fine-tuning had minimal vs. substantial gains.
- The paper is awkwardly organized, such as “limitations” abruptly inserted before the main contributions are outlined.
- “The costs and error of existing methods is the single most important bottleneck in the empirical legal studies pipeline.” (39-40) is vague and needs a citation.
- Despite the significant contribution of the dataset and language modeling performance, there is little methodological novelty to their approach. This begs the question of why ICLR is the appropriate venue for this work.

**Questions:**

- What process did you use to determine the relative difficulty of these tasks and whether they are comprehensive wrt the legal activities of humans? For instance the paper would benefit significantly from a legal expert's classification of the tasks by difficulty, followed by an analysis of Lawma according to those tasks
- What does "mixed answer" mean in Appendix G?
- Will you release the model weights and code?
- Is there any transferability across tasks—i.e., if you train a model on a subset of the task, how does it perform on the held-out tasks? Do you anticipate that in the future all subtasks of legal reasoning need to be spelled out, or is there a critical mass of legal subtasks that accrue towards "legal AGI" ?

---

> ### Author Response · Authors · 2024-11-26
>
> Thank you for your thoughtful review.
>
> > There is little exploration of the relative difficulty of the different types of tasks they introduce—it would be useful to know more about which tasks that GPT-4 outperformed on, and where fine-tuning had minimal vs. substantial gains.
>
> Fortunately, the Court of Appeals database offers one measure of relative task difficulty for human coders: intercoder agreement. We discuss how intercoder agreement compares with Lawma accuracy in Appendix C. Our findings speak to the non-trivial nature of the task suite as a classification benchmark: Lawma is far from the intercoder agreement rate for most tasks, see the newly added Figure 10. The linear fit in Figure 10 has slope 1.08 and intercept -15, indicating that Lawma is on average 15 accuracy points below the intercoder agreement rate *irrespective* of intercoder agreement. But there is large variability across tasks. In fact, there are many tasks (e.g, “CIRCUIT”) with perfect intercoder agreement rate, no class imbalance, and yet for which Lawma is far from the agreement rate.
>
> Regarding what tasks GPT-4 outperformed Lawma, it is the following 7/260 tasks: songer_const1, songer_summary, sc_threejudgefdc, songer_casetyp1_1-3-3, songer_st_v_st, sc_respondentstate, songer_casetyp1_2-3-2. These tasks do not share much in common, are neither particularly easy nor challenging, and there are many similar tasks to those 7 (e.g., songer_const2, sc_petitionerstate, songer_casetyp1_1-3-2) for which Lawma does outperform GPT-4.
>
> We find that fine-tuning leads to large performance gains across all tasks. Lawma 8B improves upon the performance of Llama 3 8B Instruct on average by 37 accuracy points, with the 5th percentile of improvement being 10 accuracy points. The tasks with lowest improvements tend to be those associated with finding specific case issues (songer_casetyp*), or the nature of the appellant (songer_appel*) or respondent (songer_respond*). These tasks are particularly challenging because they are highly specific and not much training data is available.
>
> > [Are the proposed tasks and their difficulty] comprehensive wrt the legal activities of humans?
>
> The proposed tasks are derived directly from the Supreme Court Database and the Songer Court of Appeals Database, the two most widely used resources in empirical legal research. Our proposed tasks are therefore highly representative of the types of annotation tasks that concern empirical legal scholarship.
>
> > “The costs and error of existing methods is the single most important bottleneck in the empirical legal studies pipeline.” (39-40) is vague and needs a citation.
>
> Michael A. Livermore and Daniel N. Rockmore. Law as Data: Computation, Text, & the Future of Legal Analysis. Santa Fe Institute Press, 2019.
>
> > What does "mixed answer" mean in Appendix G?
>
> It means that the Court considered the question but gave a "mixed" answer - for example, when the Court supported the respondent in part and supported the appellant in part, or if two issues treated separately by the court both fell within the area covered by one question and the court answered one question affirmatively and one negatively.
>
> > Is there any transferability across tasks—i.e., if you train a model on a subset of the task, how does it perform on the held-out tasks?
>
> Yes, we observe transferability across tasks, see Section 4.4 and in particular Figure 9. Specifically, fine-tuning only on the Court of Appeals tasks improves mean accuracy on the Supreme Court tasks by up to 18.8% accuracy points.
>
> > Do you anticipate that in the future all subtasks of legal reasoning need to be spelled out, or is there a critical mass of legal subtasks that accrue towards "legal AGI"?
>
> We can only confidently speak about the current state of affairs. We observe that fine-tuning only on the Court of Appeals database results in a mean case accuracy of 51.6%, compared to 82.4% for Lawma 8B (Section 4.4). That is, not fine-tuning on Supreme Court cases results in a 30.9 accuracy points drop in performance. Our results highlight the importance of fine-tuning precisely on the target tasks of interest.
>
> Fortunately, hundreds of labelled examples are often sufficient to obtain substantial performance gains (Section 4.3). Therefore, our recommendation for legal scholars is at present time the following: obtain a few hundred labeled examples using human annotators, fine-tune an open-weights model, and use the fine-tuned model to annotate the remaining cases.

---

> > ### Author Response · Authors · 2024-11-26
> >
> > > Will you release the model weights and code?
> >
> > The benchmark, fine-tuning dataset, model weights, and all code used to construct CaselawQA and fine-tune the Lawma models are publicly available; however, they cannot be directly linked here to preserve anonymity.
> >
> > > Despite the significant contribution of the dataset and language modeling performance, there is little methodological novelty to their approach. This begs the question of why ICLR is the appropriate venue for this work.
> >
> > Please note that our submission is in the "Datasets and Benchmarks" primary area of ICLR. We nonetheless believe that showing how reasonably standard methodological choices lead to significant performance improvements is a valuable contribution in its own right. Beyond its clear practical relevance for legal scholars, it also serves as a strong baseline for future research in this critically understudied application domain.

---

### Official Review · Reviewer_2w3m · 2024-11-03

**Soundness:** 4
**Presentation:** 4
**Contribution:** 3
**Rating:** 8
**Confidence:** 4

**Summary:**

This paper produces a novel dataset, CaselawQA, of broad-ranging text classification tasks, based on the annotations of two US Supreme Court and Court of Appeals datasets (for a wide range of issues, everything from whether the defense attorney was a legal aid or public defender to whether a previous precedent was being overturned) and then investigates the success of current LLMs on these tasks. They find that prompting typical LLMs produces very poor results, often below a most frequent answer baseline, prompting GPT-4 or Llama 3 70B produces moderately good results, but that much better results can be produced by using a Llama 3 8B model fine-tuned for the various tasks at issue here. The result is not only a useful new benchmark but a strong result showing that in various distant-from-the-web domains, a fine-tuned small LLM can perform much better than a state-of-the-art huge LLM.

**Strengths:**

The strengths include:
- Carefully and thoroughly done experimentation. Even when I disagreed with the choices they made for how to measure things in the main paper, the way I would have done things is usually available in the lengthy appendices. The paper reads as comprehensive, not rushed.
- A valuable new benchmark dataset, with nearly all new tasks rather than just collating existing tasks.
- The tasks of the dataset are derived from a pre-existing database by programmatic means. This is a strength since they are annotations that have been built up by lawyers and political scientists and so have ecological validity, but a weakness in that they were in a sense pre-existing rather than this being a major contribution of new labeling.
- The paper is clearly and well written.

**Weaknesses:**

- The tasks of the dataset are derived from a pre-existing database by programmatic means. This is a strength since they are annotations that have been built up by lawyers and political scientists and so have ecological validity, but a weakness in that they were in a sense pre-existing rather than this being a major contribution of new labeling.
- The paper is not very original: There is no new machine learning, there are several pre-existing benchmark legal datasets to which this adds another one, and not with a new type of task (all of them have many text classification tasks), and the central result that fine-tuning can outperform prompting has appeared in many places (as well as sometimes the opposite; it depends on various factors including model and data scale and the distance of the tasks from what appears in the pre-training and post-training data). E.g., https://www.semanticscholar.org/paper/Prompt-Engineering-or-Fine-Tuning-A-Case-Study-on-Trad-Chehab/505e4a7bedadab7f6de006c3c1e1144e272f4695, https://arxiv.org/abs/2408.01346, https://arxiv.org/abs/2402.17193, https://reglab.github.io/racialcovenants/ vs. the opposite in https://arxiv.org/abs/2309.01715 .

**Questions:**

[none of these are important fundamental questions wrt the paper]

line 129: Would not it actually be useful to show performance on LegalBench? It would be a useful test of transfer, beyond section xx, and give people a better indication of how useful Lawma would be in general for legal tasks with/without doing further fine-tuning?

line 246: I think this form of data sampling is questionable. It seems particularly questionable for binary tasks, since it ends up making the two classes balanced, which is the easiest case. Historically, it has usually been argued that text classification should be done as an unbalanced task, because that is the setting that has ecological validity, and the falsely high numbers that accuracy then gives can be dealt with by not using accuracy but macro F1 for evaluation. Of course, I saw that you have all the other settings in the appendix, so I'm not unhappy, just questioning the choice of setting for the main paper.

line 290: Similarly, here, using micro-averaging not macro-averaging seems questionable, but you provide the opposite in the appendix.

line 411: While this graph is useful, the single axis for FLOPs is really unsystematically mixing scaling training data size and model size in a way that I think can cause as much fog as light. This isn't like a chart of Chinchilla-optimally scaled models for which we are comparing performance for different amounts of compute. Rather, if I have everything right (the paper doesn't give the details), the 6 Pythia models on the left are all trained on the same amount of data but are progressively larger models. You might conclude that models larger 410M make little difference for the benchmark here. Conversely, the 3 models 2nd, 3rd, and 4th from the right (Pythia 6.9B, Llama 2 7B, and Llama 3 8B) are all approximately the same size but differ by scaling the training the scaling data (and by instruction post-training of Llama 3 70B and Llama 3 perhaps just being better done than Llama 2). These results indicate that more pre-training data still really helps (well, at least very clearly on the Supreme Court tasks). The rightmost 2 data points return to a comparison of model size (it doesn't help much at large sizes). This suggests that around line 426 that suggesting more, better pre-training data is probably also a good source for improvements. Though I think all the big LLM companies are increasingly aware of this.

Figure 8: Given that most of the curves are still point up steeply, it would be lovely to also see results for 2500 train examples. But I realize you've already expended a lot of H100 hours on this paper. On the other hand, it seems like the graphs would be closer to a logarithmic scale and the steep curve between the first two points would be avoided if you also included points for 25 main examples, and that would cost much less to add.


Typos:

111 interests > interest;
317 were > where;
340 unfeasible > infeasible;
343 compared > compared to;
Lots of things need capitalization in the references, e.g. on line 546.

---

> ### Author Response · Authors · 2024-11-26
>
> Thank you for the thoughtful, detailed, and positive review.
>
> > the central result that fine-tuning can outperform prompting has appeared in many places
>
> We believe that our results are highly interesting and consequential not because fine-tuning leads to performance improvements, but rather because we show that specializing a “small” open-weights model substantially outperforms the much larger GPT-4 model, which is what legal scholars currently tend to rely on when considering computational annotation of Court opinions. This is our central result.
>
> > there are several pre-existing benchmark legal datasets to which this adds another one, and not with a new type of task (all of them have many text classification tasks)
>
> CaselawQA focuses on the annotation of entire Court opinions, which is of critical interest for empirical legal scholars, since the costs and error of existing methods is the single most important bottleneck in the empirical legal studies pipeline. We believe that the scale and richness of CaselawQA, containing almost all variables of the Supreme Court Database and U.S. Court of Appeals Database, makes it a valuable addition to the current ecosystem of legal benchmarks.
>
> > Would not it actually be useful to show performance on LegalBench? It would be a useful test of transfer, beyond section xx, and give people a better indication of how useful Lawma would be in general for legal tasks with/without doing further fine-tuning?
>
> We study how much task-specific fine-tuning might generalize in Section 4.4 and in particular Figure 9. For LegalBench, we would expect even more stark results, as most LegalBench tasks differ substantially from those considered in our work. Put shortly, we do not recommend using the Lawma models for tasks beyond those considered in our work. Our recommendation is to always specialize models for the specific legal tasks they are intended to perform. Not doing so needlessly leaves a lot of performance on the table. Fortunately, we show that hundreds of labelled examples are often sufficient to obtain large performance gains, at least in many annotation tasks (Section 4.3).
>
> Our results suggest the viability of the following strategy for large-scale legal annotation: to obtain a few hundred labeled examples using human annotators, fine-tune an open-weights model, and use the fine-tuned model to annotate the remaining cases at scale.
>
> > [Regarding Figure 7] These results indicate that more pre-training data still really helps (well, at least very clearly on the Supreme Court tasks).
>
> We agree that pre-training on more data might still help substantially, at least for the Supreme Court tasks, whereas simply scaling model size might not. We will make this point more clear.

---

> ### Comment · Reviewer_2w3m · 2024-11-28
> **Thanks!**
>
> Thanks for your follow up to my comments!

---

### Meta-Review · Area_Chair_WLu5 · 2024-12-24

**Metareview:**

This work presents a classification dataset in the legal domain. Experiments demonstrate that a Llama model fine-tuned on in-domain data outperforms the GPT-4 model.

All reviewers recommended acceptance. However, they also highlighted several weaknesses, summarized below:

- The dataset primarily leverages existing annotations from two legal datasets, which limits the scope of new contributions.
- Prior research has shown that fine-tuned, domain-specific models often outperform general-purpose LLMs, reducing the methodological novelty and in-depth insights of this work.

Taking into account the overall reviews and discussions, we are pleased to recommend acceptance of the paper as a poster presentation.

**Additional Comments On Reviewer Discussion:**

No new points raised after rebuttal.

---

### Decision · Program_Chairs · 2025-01-22

Accept (Poster)